# ZAK activation at the collided ribosome

Vienna L. Huso[1,2,5], Shuangshuang Niu[3,5], Marco A. Catipovic[1,2], James A. Saba[1,2], Timo Denk[3], Eugene Park[1,2], Jingdong Cheng[4], Otto Berninghausen[3], Thomas Becker[3], Rachel Green[1,2 ✉] & Roland Beckmann[3 ✉]

Ribosome collisions activate the ribotoxic stress response mediated by the MAP3K ZAK, which in turn regulates cell-fate consequences through downstream phosphorylation of the MAPKs p38 and JNK[1]. Despite the critical role of ZAK during cellular stress, a mechanistic and structural understanding of ZAK–ribosome interactions and how these lead to activation remain elusive. Here we combine biochemistry and cryo-electron microscopy to discover distinct ZAK–ribosome interactions required for constitutive recruitment and for activation. We find that upon induction of ribosome collisions, interactions between ZAK and the ribosomal protein RACK1 enable its activation by dimerization of its SAM domains at the collision interface. Furthermore, we discover how this process is negatively regulated by the ribosome-binding protein SERBP1 to prevent constitutive ZAK activation. Characterization of novel SAM variants as well as a known pathogenic variant of the SAM domain of ZAK supports a key role of the SAM domain in regulating kinase activity on and off the ribosome, with some mutants bypassing the ribosome requirement for ZAK activation. Collectively, our data provide a mechanistic blueprint of the kinase activity of ZAK at the collided ribosome interface.

The ribosome translates mRNA into protein, often with multiple ribosomes on a given mRNA called polysomes. Ribosomes are also essential sensors of cellular stress and can alert the cell of nutrient deprivation[2], damage to mRNAs and chemical insults that directly target and damage ribosomes[3–5]. Such cellular stresses cause ribosomes to stall on problematic mRNA, resulting in the lagging ribosome colliding with the stalled ribosome. These ribosome collisions are a key signal to activate both quality control pathways and broad stress signalling responses[4,6].

For one of these signalling pathways, the ribotoxic stress response (RSR), the mitogen-activated protein kinase kinase kinase (MAP3K) ZAKα (referred to hereafter as ZAK) has a central role in orchestrating the RSR[7–9]. Previous studies have demonstrated that ZAK interacts constitutively with ribosomes during unstressed conditions but becomes activated (via autophosphorylation) and is released from the ribosome upon cellular stresses that impair translation[1,10]. Although previous studies have argued that ribosome collisions are key determinants of ZAK activation[1], other studies have suggested that both colliding and individually stalled ribosomes may be potent triggers[10,11]. In either case, ZAK activates the stress-activated protein kinases (SAPKs) p38 and JNK, leading to cell cycle arrest and/or apoptosis[12–14].

Kinase regulation often requires scaffold proteins[15], and ZAK has been shown to interact with 14-3-3 proteins off the ribosome in a canonical phosphorylation-dependent manner downstream of activation[10,12]. Another scaffold is the receptor for activated C-kinase (RACK1), a conserved eukaryotic ribosomal protein on the head of the 40S subunit, which was first characterized for PKC activation and later argued to be a signalling hub for kinases including JNK[16–19]. Of note,

RACK1 resides exactly at the collision interface and has a critical role in collision-mediated quality control events[20–24], hinting that RACK1 could scaffold ZAK on the ribosome.

Although much is known about ZAK signalling and its downstream consequences for cell fate[12,13], the molecular determinants of the interaction of ZAK with the ribosome and an understanding of how these interactions mediate activation have remained enigmatic. Here we elucidate how ZAK is recruited to ribosomes, both in its basal state and in induced stress conditions, and show how collision-specific interactions organized on RACK1 mediate ZAK activation.

## ZAK enrichment on ribosomes

A single mammalian cell contains approximately 5 million ribosomes, and the ratio of ZAK to ribosomes is estimated at approximately 1:100 based on copy numbers measured across various cell types[25,26]. Therefore, we turned to overexpression of N-terminally tagged ZAK in HEK293T cells to enrich ZAK-bound ribosomes. We characterized ribosome binding of wild-type (WT) ZAK and monitored its phosphorylation status by Phos-tag immunoblotting after sucrose gradient fractionation. Overexpressed WT ZAK was found in the top fractions of the gradient and migrated on western blot (Phos-tag) at a molecular weight of approximately 250 kDa, consistent with fully phosphorylated (P) 'activated' protein[1,12] (Fig. 1a); in addition, JNK-P levels were increased under basal conditions (Extended Data Fig. 1a). These observations reflect activation of ZAK and the RSR upon ZAK overexpression.

We next overexpressed a kinase inactive ZAK with mutations in the activation loop: T161A/S165A[27]. Despite its strong overexpression, this

[1]Department of Molecular Biology and Genetics, Johns Hopkins University School of Medicine, Baltimore, MD, USA. [2]Howard Hughes Medical Institute, Johns Hopkins University School of Medicine, Baltimore, MD, USA. [3]Department of Biochemistry, Gene Center, University of Munich, Munich, Germany. [4]Minhang Hospital & Institutes of Biomedical Sciences, Shanghai Key Laboratory of Medical Epigenetics, International Co-laboratory of Medical Epigenetics and Metabolism, Fudan University, Shanghai, China. [5]These authors contributed equally: Vienna L. Huso, Shuangshuang Niu. ✉e-mail: ragreen@jhmi.edu; beckmann@genzentrum.lmu.de

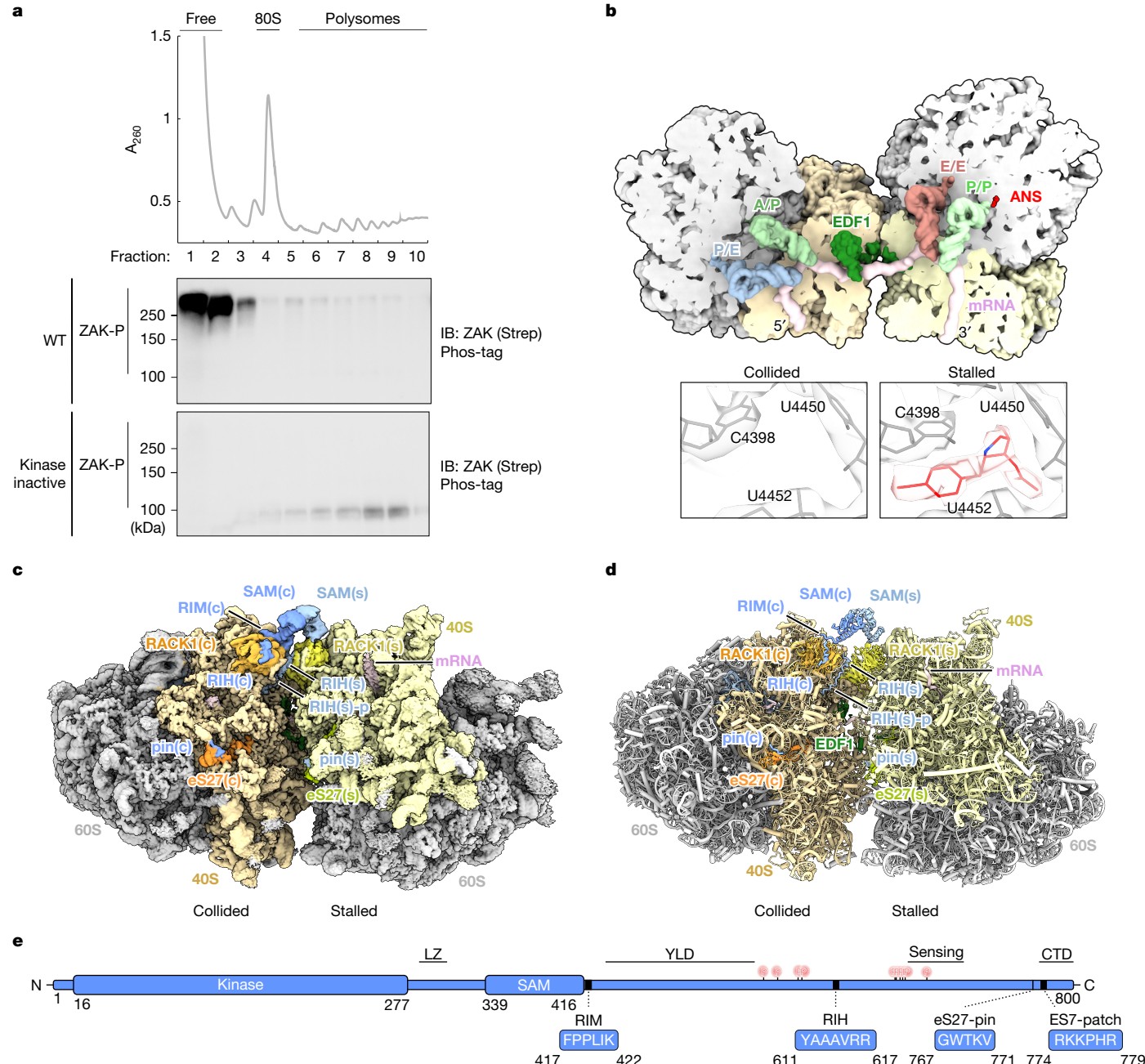

**Fig. 1 | Cryo-EM structure of ZAK bound to a colliding disome. a**, Polysome profile (top) and immunoblots of a Phos-tag gel (IBs; bottom) from sucrose gradient fractions of HEK293T cells transfected with Strep-tagged WT and kinase inactive ZAK(T161A/S165A) expressed from a complete CMV promoter plasmid. Polysome profile from WT transfection is shown. **b**, Cut top view on the ZAK–disome model shown as low-pass-filtered surface, highlighting mRNA, tRNA and EDF1-binding sites. The boxed panels show a zoomed-in view on the ANS (red)-binding site of refined cryo-EM maps (transparent) of the stalled (right) 80S and collided (left) 80S. **c**,**d**, Composite cryo-EM map (**c**) and molecular model (**d**) of the ZAK–disome complex. Structural features of ZAK are indicated as SAM domain (SAM), pin, RIH, RIM and RIH-p. **e**, Schematic of the ZAK domain architecture based on AlphaFold prediction: N-terminal kinase domain, leucine zipper (LZ) region, SAM, YLD, sensing region and the CTD. Residues relevant for ribosome interaction as identified in this study are depicted in blue boxes. Vertical lines with P (red) indicate previously published subset of RSR phosphorylation sites (cluster 2)[12]. The blots represent at least two independent replicates (see the section 'Statistics and reproducibility'). See Supplementary Fig. 1 for source data.

variant remained deep in the polysome fractions and migrated on western blot (Phos-tag) at the expected molecular weight for unphosphorylated inactive ZAK (Fig. 1a). Neither this ZAK mutant (T161A/S165A) nor another variant with a mutated ATP-binding site (K45M) led to increased levels of JNK-P when overexpressed (Extended Data Fig. 1a). Because these variants are not released from ribosomes in response to collisions[12], we decided to use them as tools for structural studies targeting ribosome-bound ZAK.

## ZAK binds to the collided ribosome interface

We performed native pull-downs of inactive tagged ZAK variants from Expi293F cells either without or with low-dose anisomycin (ANS) to induce ribosome collisions[1,28] (Extended Data Fig. 1b). We found that ribosomes co-enriched with all ZAK variants and independent of ANS addition (Extended Data Fig. 1b, left panel). We performed single-particle cryo-electron microscopy (cryo-EM) analysis of treated and

untreated samples (Extended Data Figs. 2 and 3a). Three-dimensional classification showed large classes of stable disomes in the ANS-treated sample that were absent in the untreated sample, with extra density for ZAK at the collision interface. The most abundant disome class was refined to generate a composite ZAK-bound disome map (Fig. 1b,c and Extended Data Fig. 2; see Methods for details). A molecular model was generated based on existing human 80S structures[29,30] and guided by AlphaFold multimer predictions for different regions of ZAK[31–33] (Fig. 1d, Extended Data Table 1 and Extended Data Figs. 4a–c and 5a–e).

The ZAK-bound disome was in a typical arrangement as described before for human disomes[34]. The stalled ribosome contained P/P-site and E/E-site tRNAs, and, as expected, ANS was visualized in the peptidyl-transferase centre. The collided ribosome contained hybrid A/P–P/E-state tRNAs but no density for ANS (Fig. 1b). EDF1 was positioned on the collided ribosome as described[21], with the N-terminal domain now resolved and reaching over to the stalled ribosome (similar to its homologue Mbf1 in yeast) to interact with ribosomal protein eS26 (ref. 35) (Fig. 1b and Extended Data Fig. 5f,g).

## ZAK contacts the disome at multiple sites

ZAK consists of an N-terminal kinase domain (residues 16–277), leucine zipper region (residues 280–326), sterile-α motif (SAM) (residues 339–416) and a computationally predicted YEATS-like domain (YLD; residues 433–551)[36] (Fig. 1e and Extended Data Fig. 4b). The C-terminal 100 amino acids (residues 700–800) are critical for ribosome binding and activation (ribosome-binding region (RBR)), with the last 27 amino acids enriched in positively charged residues (residues 774–800; C-terminal domain (CTD)) being of particular importance[1,10] (Fig. 1e).

In the refined structure, we identified density for different parts of ZAK. The largest density forms a bridge connecting RACK1 of the stalled ribosome, RACK1(s), with RACK1 of the collided ribosome, RACK1(c), and represents a dimer formed by two ZAK SAM domains (Extended Data Fig. 4c–e). Each SAM domain binds RACK1 with a small motif immediately downstream of the SAM domain, termed the RACK1-interacting motif (RIM; residues 417–422; Fig. 1c–e and Extended Data Figs. 4b and 5a–e). In addition, further downstream, a short α-helix serves as a second RACK1 interaction motif, termed the RACK1-interacting helix (RIH; residues 611–617; Fig. 1c–e and Extended Data Figs. 4b and 5a–e). Of note, the RIH on RACK1(s) features an additional peptide that reaches across the collision interface to interact with the collided ribosome, termed the RIH-peptide (RIH-p; residues 618–630; Extended Data Fig. 4b,e). Finally, on each ribosome, another short peptide (residues 767–771) interacts with ribosomal protein eS27 (eS27(s) and eS27(c); Fig. 1c–e and Extended Data Figs. 4b and 5a–e), termed the eS27-pin (pin). Consurf analysis[37] showed high conservation of residues present in each motif (Extended Data Fig. 1c). Although the remaining regions of ZAK are dynamic and not resolved, additional globular density emerges from the SAM dimers at low contour levels and when low-pass filtered to approximately 30 Å matches in overall size and shape a dimer of the leucine-zipper and kinase domains (Extended Data Fig. 2).

Together, the structure reveals how the disome facilitates recognition and interaction between the SAM domains of ZAK. We hypothesize that this structure represents a pre-activation state with ZAK primed for activation. In the following sections, we investigated the functions of each ZAK structural unit observed on the colliding disome.

## ZAK C terminus binds to the 40S subunit

We first focused on the C terminus of ZAK. The density for the pin (residues G767–V771) was visible on all ZAK-bound 80S ribosomes, regardless of treatment or collision state (Extended Data Fig. 5a–e). These classes comprised both translating (bound to tRNAs and mRNAs) and hibernating (bound to eEF2–SERBP1) 80S (Extended Data Figs. 2,

3a and 5b,c), indicating that ZAK is guided by the pin to ribosomes independent of translation state. In vitro binding assays further confirmed that GST-tagged RBR bound 40S subunits and 80S ribosomes but not 60S subunits (Extended Data Fig. 1d,e). Cryo-EM reconstructions of reconstituted GST–RBR–40S complexes revealed well-resolved (2.3 Å) extra density on eS27, validating the pin interaction observed in native ZAK immunoprecipitations as a general 40S–80S recruitment interaction and allowing interpretation at the molecular (side chain) level (Extended Data Fig. 5d, second row). The pin is anchored to eS27 via W768, which intercalates between R80 and K36 of eS27 with further contacts contributed by adjacent residues (T769, K770 and V771) of ZAK (Fig. 2b and Extended Data Figs. 5d and 6a).

Of note, the pin places the highly charged CTD in close proximity to the 18S rRNA expansion segment ES7 (Fig. 2c, dashed line). Although not visualized, multimodal electrostatic interactions between the negatively charged rRNA backbone and the positively charged CTD probably contribute to ZAK binding. Furthermore, only in the disome structure the N-terminal region flanking the pin is in close contact to ES6c of the collided ribosome (Fig. 2c and Extended Data Fig. 6b).

We next generated a system to characterize the effects of ZAK mutations on ribosome binding and activation by expressing ZAK at endogenous levels using an expression plasmid with a partial CMV promoter (Extended Data Fig. 1a). *ZAK*-knockout cells carrying the WT ZAK expression construct revealed ZAK and JNK activation upon ANS treatment resembling endogenous ZAK behaviour (Extended Data Fig. 1a). We followed ZAK binding to ribosomes using analytical sucrose gradients. In WT cells, ZAK distributes across the free and polysome fractions and is increased at the top of the gradient upon ANS treatment, consistent with ribosome dissociation following activation (Extended Data Fig. 1f). These data match previous observations[1,10], although more clearly reveal unbound ZAK. The observation of an equilibrium between ribosome-bound and unbound ZAK populations is consistent with a role in surveying ribosome complexes in the cell.

We next generated a point mutation of the eS27-interacting residue (W768A; pin$_{mut}$; Fig. 2d) and observed substantially reduced ZAK binding to polysomes independent of ANS treatment (Fig. 2e). We also predicted residues probably bound to ES7 based on the location and charge and generated a four-alanine patch mutant (R774A, K775A, K776A and R779A; ES7-patch$_{mut}$), which also revealed substantial loss of ZAK binding to polysomes (Fig. 2d,e). Mutating all five residues together (pin$_{mut}$ + ES7-patch$_{mut}$) resulted in near complete loss of ribosome binding (Fig. 2e), which correlated directly with losses in ZAK activation and JNK phosphorylation (Fig. 2f). Together, the C-terminal interactions of ZAK with eS27 and ES7 contribute to overall affinity of ZAK for ribosomes and serve as a prerequisite for activation upon collision.

## Collision-specific rRNA interaction

We further explored the interaction of ZAK with 18S rRNA using crosslinking and immunoprecipitation sequencing (CLIP-seq). We transfected *ZAK*-knockout cells with overexpressed kinase inactive ZAK(T161A/S165A), a C-terminal truncation mutant (residues 1–649) defective for ribosome binding[1] or a mock transfection control (ZAK knockout; Fig. 2g). We compared all samples to the C-terminal truncation mutant. We observed greater than twofold enrichment of ZAK footprints mapping to ES7 (18S; bases 1117–1195) in both untreated conditions (Fig. 2g, top) and with ANS (Fig. 2g, bottom). These observations establish that the C terminus of ZAK indeed forms contacts with the ribosome via ES7 independent of collisions, indicative of a 'sampling' mode.

A second footprint region emerged for 18S rRNA bases 710–766 corresponding to ES6b and ES6c (ES6b/c; Fig. 2g, bottom) only upon treatment with ANS. These data are consistent with our structural observations (Extended Data Fig. 6b) and suggest that portions of the

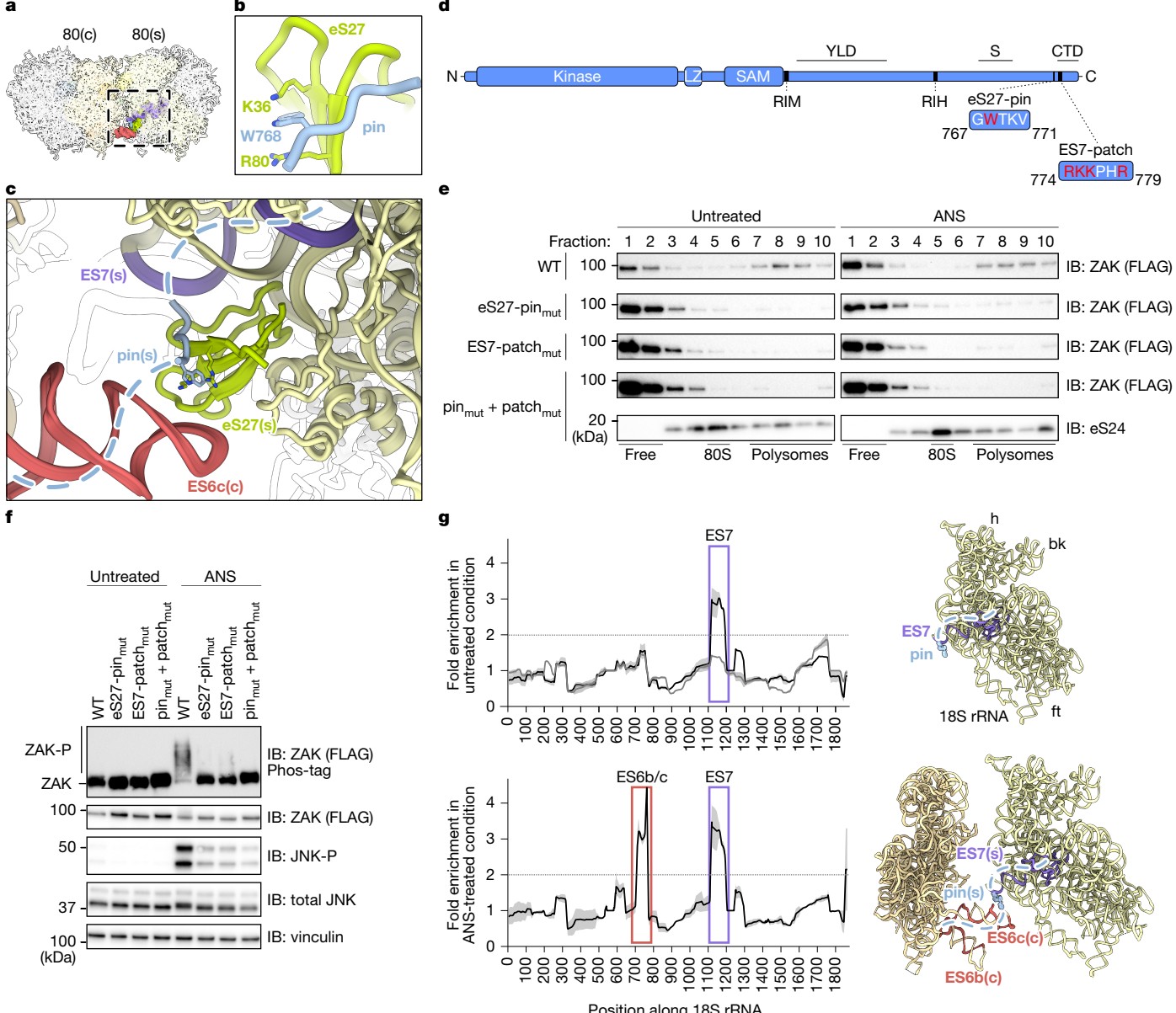

**Fig. 2 | The C terminus of ZAK mediates ribosome binding and collision-specific rRNA interaction. a**, Top view on the molecular model of the ZAK–disome. The box highlights the site of the pin on the stalled ribosome, comprising the collision interface formed by ES7 and eS27 (both on stalled) and ES6c on the collided ribosome. **b**, Molecular model of the pin within the ZAK RBR, bound to eS27 on the stalled 80S (see also Extended Data Figs. 5a–e and 6a). **c**, Zoomed-in view on the collision interface as outlined in panel **a**. The dashed line indicates regions flanking the pin. **d**, Schematic of the ZAK domain architecture, with important residues in the RBR highlighted. Residues marked in red were targeted for mutational analysis. S, sensing domain. **e**, Immunoblots of sucrose gradient fractions collected from HEK293T *ZAK*-knockout cells transfected with partial CMV promoter plasmids expressing various N-terminal FLAG ZAK constructs at endogenous levels. Immunoblots were probed for FLAG to visualize ZAK. A representative eS24 blot is shown (Extended Data Fig. 3b).

**f**, Immunoblots of total lysate from gradients shown in panel **e**. **g**, CLIP-seq of FLAG-tagged ZAK(T161A/S165A) expressed from the full CMV promoter in HEK293T *ZAK*-knockout cells and associated ribosomal RNA (black lines) in untreated and ANS-treated conditions. Sequence reads were mapped to 18S rRNA and normalized to the non-ribosome binding 1–649 amino acid C-terminal truncation control. The shading indicates standard error of the mean from two biological replicates. *ZAK*-knockout cells are shown as a grey line in the untreated blot. The horizontal dotted line represents twofold enrichment. Reads from CLIP-seq were mapped to the 18S rRNA (monosome for untreated; disome for ANS-treated) and matched to ES7 (purple) and ES6b/c (red). The position of the pin of ZAK is highlighted with blue spheres. The blots represent at least two independent replicates (see the section 'Statistics and reproducibility'). See Supplementary Fig. 2 for source data. bk, 40S beak; ft, 40S foot; h, 40S head.

C terminus of ZAK engage ES6c (ES6-patch) on the collided ribosome (ES6c(c)). Additional CLIP experiments performed with WT and kinase inactive ZAK(T161A/S165A) expressed at endogenous levels showed similar strong enrichment for ES6b/c upon ANS treatment (Extended Data Fig. 6c). No read enrichment was observed near helix 14, a region previously suggested to interact with ZAK[10]. Similarly, none of the ZAK constructs revealed CLIP signal on the 28S rRNA of the 60S subunit

relative to controls consistent with the ZAK–ribosome interaction being mediated by the 40S subunit (Extended Data Fig. 6d). Finally, metagene analysis revealed enrichment of ZAK CLIP reads in the open reading frames of mRNAs for all full-length ZAK samples. This observation probably reflects ZAK interactions with mRNA on the ribosome due to its proximity to mRNA entry and exit sites between 40S body and head (Extended Data Fig. 6e).

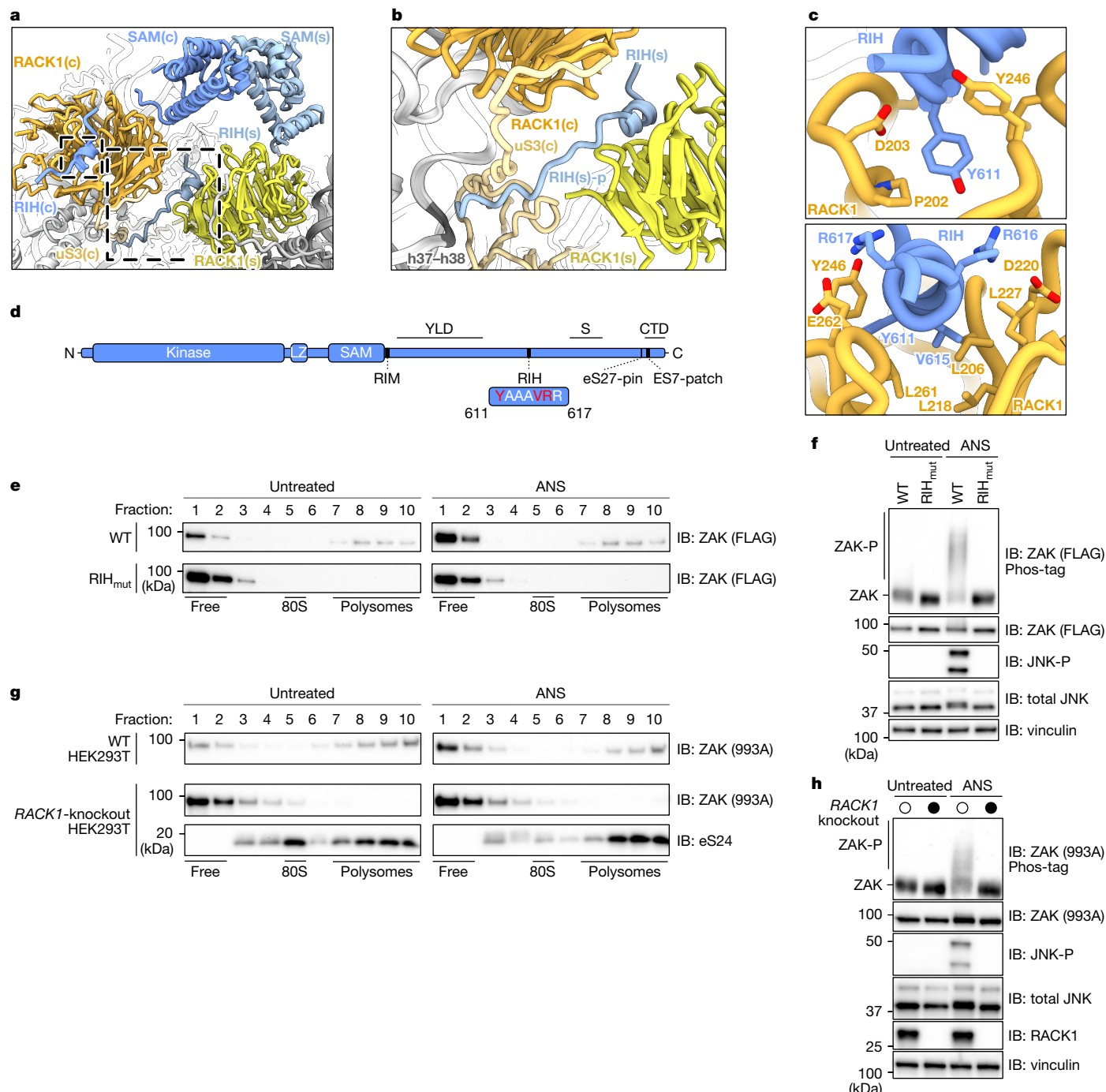

**Fig. 3 | ZAK interaction with RACK1 via RIH mediates ribosome binding.**
**a**, Zoomed-in view of the molecular model of the ZAK–disome focusing on the RACK1 collision interface. The boxes highlight the position of the RIH and the RIH-p. **b**, Zoomed-in view focusing on the RIH of the stalled 80S (RIH(s)) and the RIH-p extending towards the disome interface to form contacts with uS3 and h37–h38 of the collided 80S. **c**, Two views highlighting the interactions of RIH Y611 and V615 with RACK1. Additional contacts may be formed by R616 and R617 of ZAK with D220 and E262 of RACK1. **d**, Schematic of the ZAK domain architecture with RIH highlighted. Residues marked in red were targeted for mutational analysis. **e**, Immunoblots of sucrose gradient fractions collected from HEK293T *ZAK*-knockout cells transfected with partial CMV promoter plasmids expressing various N-terminal FLAG ZAK constructs at endogenous levels. Immunoblots were probed for FLAG to visualize ZAK (see Extended Data Fig. 3c for eS24 blots). **f**, Immunoblots of total lysate from gradients shown in panel **e**. **g**, Immunoblots of sucrose gradient fractions collected from HEK293T WT or *RACK1*-knockout cells. Immunoblots were probed for endogenous ZAK (993A antibody), and the eS24 blot is shown in Extended Data Fig. 3d. **h**, Immunoblots of total lysate from gradients shown in panel **g**. Blots represent at least two independent replicates (see the section 'Statistics and reproducibility'). See Supplementary Figs. 3 and 4 for source data.

## RIH mediates ribosome binding

Next, we focused on the RIH, a short α-helical motif (residues 611–617) anchoring ZAK at the disome interface between blade 5 and blade 6 of RACK1 (Fig. 3a and Extended Data Fig. 4f). Our well-resolved maps (below 3 Å in the RACK1 region) revealed Y611, pointing towards the backbone loop of RACK1 (residues 242–248), and V615, packing against a hydrophobic surface of RACK1 (L206, L218 and L261) as the main interacting RIH residues (Fig. 3c, top, and Extended Data Fig. 5, third row). Of note, although being present on both RACK1 proteins, only the

RIH on the stalled 80S features an additional short peptide (residues 618–630; RIH-p) that extends towards the collided 40S. This peptide potentially interacts with the C terminus of uS3 and rRNA at the junction of helices h37 and h38 (h37–h38) on the colliding ribosome (Fig. 3b and Extended Data Fig. 4e). Similar to the pin, we observed density for the RIH in almost all subclasses of 80S ribosomes regardless of ANS treatment (Extended Data Fig. 5b,c), indicating that the RIH constitutes another general ribosome-binding motif.

We generated a RIH ZAK mutant (Y611, V615 and R616A; RIH$_{mut}$; Fig. 3d) and observed complete loss of binding to polysomes in untreated and ANS-treated conditions (Fig. 3e), and, in addition, ZAK activation by ANS was completely abolished (Fig. 3f). This loss of ribosome binding and activation was recapitulated in a *RACK1*-knockout HEK293T cell line, where endogenous ZAK migrates entirely in the free fractions (Fig. 3g) and ZAK (and JNK) phosphorylation upon ANS treatment were lost (Fig. 3h). From these data, we conclude that the RIH, similar to the pin, makes collision-independent contacts with RACK1 necessary for ZAK binding and activation.

## FPxL motif mediates collision sensitivity

We next focused on the RIM located immediately downstream of the SAM domain. The RIM was exclusively observed on collided ribosomes (Fig. 4a) and consists of a short peptide motif (FPPLIK; residues 417–422) that stretches over RACK1 blades 2 and 3 (Extended Data Fig. 4f; identical in both the stalled and the colliding ribosome). As revealed by our sub-3 Å maps in this region, contacts of the RIM are primarily established by F417 that caps the α5 of SAM and binds into a groove clad by Q119 and N133 of RACK1 blade 3, and by L420 that inserts into a hydrophobic pocket formed by F77, L89 and F113 between RACK1 blades 2 and 3 (Fig. 4b,c and Extended Data Fig. 5, fourth row).

Mutating the three RIM residues directly contacting RACK1 (F417, L420 and K422A; RIM$_{mut1}$; Fig. 4d) did not disrupt ZAK binding to polysomes and RIM$_{mut1}$ remained bound upon ANS treatment (Fig. 4e). We observed the same results when mutating these residues as well as two proximal prolines (F417, P418, P419, L420 and K422A; RIM$_{mut2}$; Fig. 4d,e). These data indicate that unlike the constitutive binding activity of the pin and the RIH, the RIM does not have a role in the constitutive association of ZAK with ribosomes. However, the same RIM mutants abrogated ZAK and JNK activation (Fig. 4f) and exhibited persistent ribosome binding even upon treatment with ANS (Fig. 4e). Thus, although the RIM is not critical for ZAK binding, it is strictly required for ZAK activation on collided ribosomes.

## SERBP1 negatively regulates ZAK

We observed SERBP1 density at the RIM interaction site of RACK1 in both hibernating and translating 80S classes from the ZAK pull-down (Fig. 4g). These monosome classes also revealed density for a single ZAK RIH (also on RACK1) and the pin (Extended Data Fig. 5b,c). SERBP1 is a ribosome-associated factor that binds to dormant ribosomes, usually together with eEF2 (refs. 38,39). SERBP1 has a well-characterized internal region that binds to the mRNA channel, and a recent in situ single-particle cryo-EM study on native ribosomes in human cells has revealed that the uncharacterized C terminus of SERBP1 (SERBP1-C) contacts RACK1 in dormant and translating ribosomes[40]. Our density matches those data (Extended Data Fig. 5b,c,e) and agrees with an Alpha-Fold prediction of a RACK1–SERBP1 complex. The RACK1-interacting patch of SERBP1 shows remarkable structural similarity to the ZAK–RIM and contains a consensus FPxL sequence where F404 and L407 of SERBP1 contact RACK1 (Fig. 4h). Upstream of F404, SERBP1 continues to loop over RACK1, forming additional contacts via P396 and V398 with RACK1 blades 3 and 4, respectively. Thus, our data show that ZAK and SERBP1 share a common motif for RACK1 binding, suggesting that SERBP1 could have a role in regulating ZAK activation through

direct competition. We addressed this question by monitoring RSR activation under SERBP1 knockdown conditions (with short interfering RNAs (siRNAs)) and saw increased JNK phosphorylation both in basal and activating conditions (Fig. 4i). These data are consistent with the model that SERBP1 acts as a negative regulator of ZAK, and perhaps other factors with similar motifs (see Discussion), through competition for RACK1 binding.

## RACK1 scaffolds SAM dimer on disome

The largest additional density in the ZAK–disome structure from the ANS-treated ZAK pull-down bridges the two RACK1 proteins directly adjacent to the two RIMs (Fig. 5a and Extended Data Fig. 4d,e). Although the local resolution decreased with the distance from the ribosomes, from below 3 Å close to RACK1 to about 5 Å and 8 Å at the periphery (Extended Data Fig. 2), we could position an AlphaFold model for a SAM domain dimer based on its characteristic five-helix bundle shape for each SAM domain (Fig. 5a and Extended Data Fig. 4c–e,h). The long helix α5 proceeds from the ZAK–RIM interaction like a leg scaffolding the four shorter α-helices.

In the AlphaFold model, the two ZAK SAM domains form the common asymmetric head-to-tail interface[41] (Extended Data Fig. 4c) involving a network of salt bridges formed between K394 and K387 on the stalled ribosome with D388 and D385 on the collided ribosome (Fig. 5b, right). Of note, on the stalled ribosome, we observed a rod-like extra density packed against SAM(s) at the ribosome-facing side of the helical bundle, which we assigned to an α-helix formed by residues 569–583 of ZAK ('helix'; Extended Data Fig. 4b,d,e). The two SAM domains are directly anchored to the two RACK1s via their tight RIM connection and the SAM dimer can only form when the two RACK1s are juxtaposed upon ribosomal collision. Therefore, the SAM domains could in principle monitor and read out the distance between stalled and collided ribosomes. We hypothesize that formation of the SAM dimer represents the hallmark of collision sensing by ZAK, which ultimately leads to conformational changes in the N-terminal domains of ZAK to license kinase activation, thus resembling the mechanism of other SAM-containing kinases[42,43].

## SAM interface modulates kinase activity

The known disease-associated mutation in the SAM domain (F368C) causes a hyperactive phenotype characterized by ZAK activation similar to another previously characterized constitutively active W347S mutant[10,44]. Neither mutation is located at the SAM–SAM interface but in the hydrophobic core of the domain (Fig. 5b, left). Co-immunoprecipitations with uniquely tagged ZAKs (FLAG and haemagglutinin) revealed dimerization of WT ZAK and of the W347S and F368C variants (Extended Data Fig. 7a) with modest enrichment upon ANS treatment. These data are consistent with ZAK activation producing a more stable kinase dimer and suggests that ZAK exists at least in part as a dimer in untreated conditions. We mutated residues at the SAM–SAM interface to either alanine (K/R/D→A) or reversed charge (K/R→D) and characterized ZAK activation (Fig. 5b, right, 5c). These mutations resulted in an array of activity phenotypes ranging from completely inactivating to constitutively activating, indicating that the SAM–SAM interaction has a key role in kinase activity regulation (Extended Data Fig. 7b).

We used the pathogenic F368C variant as a positive control for hyperactivity and compared it with our strongest hyperactive interface mutant (K387D) and our most inactive interface mutant (K394D; Fig. 5d). After 24 h of expression of the variants in untreated *ZAK*-knockout HEK293T cells, we observed a substantial decrease in overall ZAK levels for both F368C and K387D compared with WT (Fig. 5d, lanes 1–3), consistent with previous studies showing that ZAK is degraded downstream of activation via a phosphodegron mechanism[10,12]. By contrast, the inactive K394D mutant exhibited stable

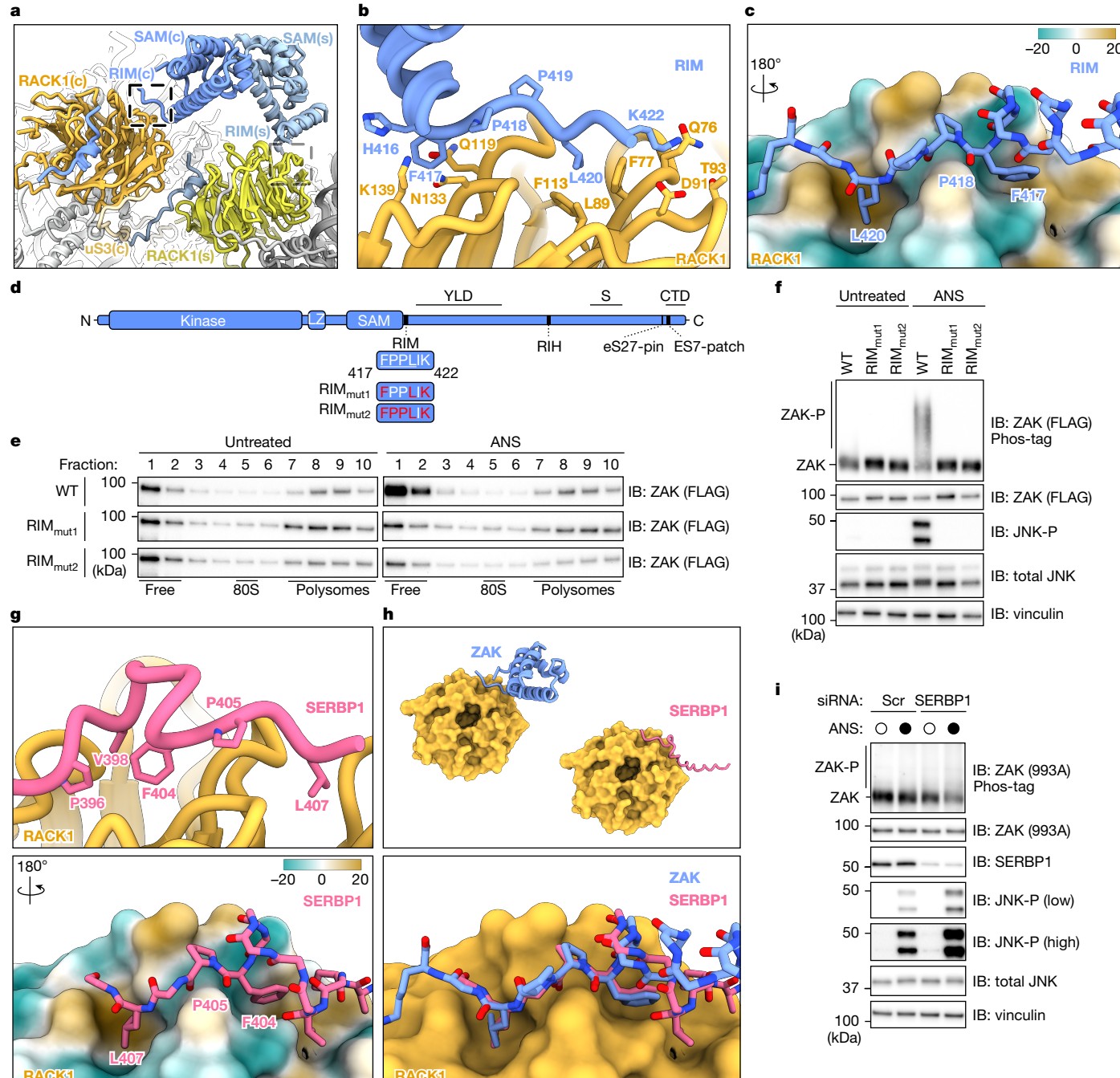

**Fig. 4 | RIM on ZAK mediates collision sensitivity and activation. a**, Zoomed-in view on the molecular model of the ZAK–disome as in Fig. 3a. The boxes highlight the position of the RIM. **b**, Zoomed-in view highlighting the interactions between RACK1 and the ZAK RIM–F417 that caps the α5 of SAM, which binds into a groove clad by Q119 and N133 of RACK1 blade 3; L420 inserts into a hydrophobic pocket formed by F77, L89 and F113 between RACK1 blades 2 and 3. In addition, K422 forms a salt bridge with T93 and a hydrogen bond to Q76 of RACK1 blade 2 and H416 of SAM α5 contacts K139 in blade 3 of RACK1. Beyond F417, the RIM merges into the last α-helix of the SAM domain (α5) that extends into the solvent space above the 40S head. **c**, A 180° rotated view of **b** with RACK1 shown as surface colour coded according to lipophilicity potential (scale bar shown). **d**, Schematic of the ZAK domain architecture with the RIM highlighted. Residues marked in red were targeted for mutational analysis. Predicted FPxL motif residues are underlined. **e**, Immunoblots of sucrose gradient fractions collected from HEK293T *ZAK*-knockout cells transfected with partial CMV promoter plasmids expressing various N-terminal FLAG ZAK constructs at endogenous levels. Immunoblots were probed for FLAG to visualize ZAK (see Extended Data Fig. 3e for eS24 blots). **f**, Immunoblots of total lysate from gradients shown in panel **e**. **g**, Zoomed-in view on the molecular model of the SERBP1 C terminus including the FPxL motif bound to RACK1. The bottom panel shows RACK1 coloured by lipophilicity potential. **h**, Comparison of the RACK1-bound ZAK RIM and the SERBP1 C terminus. The top panel shows the overall position. The bottom panel shows an overlay. **i**, Immunoblots of total lysate from HEK293T cells treated with non-targeting (Scr) or *SERBP1*-targeting (SERBP1) siRNA. The blots represent at least two independent replicates (see the section 'Statistics and reproducibility'). See Supplementary Figs. 5 and 6 for source data.

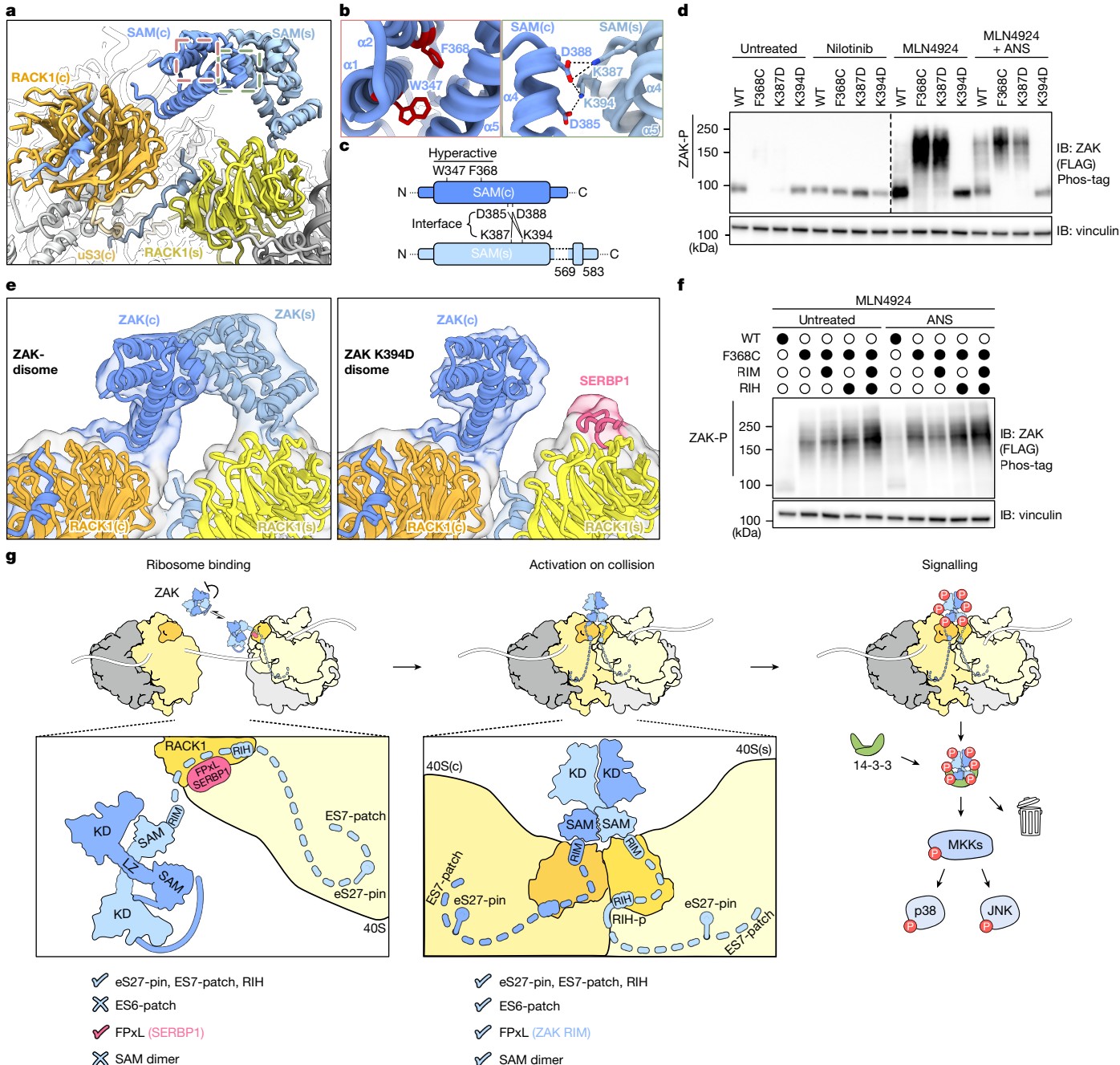

**Fig. 5 | SAM dimerization of ZAK regulates its kinase activity. a**, Zoomed-in view on the molecular model of the ZAK–disome as in Fig. 3a. The boxes highlight the position of the hyperactive phenotype mutants (red dashed box) and SAM dimer interface (green dashed box) as shown in **b**. **b**, Location of hyperactive phenotype mutants within the SAM five-helix bundle (left) and zoomed-in view on amino acids involved in a network of salt bridges (highlighted by dashed lines) within the asymmetric head-to-tail interface of the SAM dimer (right). **c**, Schematic of ZAK SAM bridge interaction with characterized interface residues highlighted. **d**, Immunoblots of total lysate from HEK293T *ZAK*-knockout cells transfected with partial CMV promoter plasmids expressing various SAM mutant N-terminal FLAG ZAK constructs at endogenous levels. **e**, Zoomed-in view on low-pass-filtered cryo-EM maps (transparent) of the ZAK–disome (left) and ZAK(K394D)–disome (right), highlighting the RACK1 collision interface with molecular models for a RACK1-bound SAM dimer or SAM monomer (on RACK1(c)) and SERBP1 (on RACK1(s)) docked. **f**, Immunoblots of total lysate from HEK293T *ZAK*-knockout cells

transfected with partial CMV promoter plasmids expressing various N-terminal FLAG ZAK constructs at endogenous levels. **g**, Model of ZAK activation at the collided ribosome. Under unstressed conditions, ZAK interacts with ribosomes via the pin, ES7-patch and RIH, whereas the FPxL motif of SERBP1 is bound to RACK1 ('Ribosome binding'). Upon ribosome collisions, the RIMs (FPxL motifs) bind to the two proximal RACK1s, the SAM domains dimerize and ZAK becomes active and autophosphorylated ('Activation on collision'). Phosphorylated ZAK is released and forms a complex with 14-3-3 scaffold proteins targeting downstream effectors in the RSR pathway ('Signalling'). The 60S subunit is in grey, the stalled 40S(s) or collided 40S(c) subunit is in yellow, the stalled RACK1 is in light orange, the collided RACK1 is in dark orange, ZAK is in light or dark blue, SERBP1 is in pink and 14-3-3 proteins are in green. The blots represent at least two independent replicates (see the section 'Statistics and reproducibility'). See Supplementary Fig. 7 for source data.

protein levels and migrated at a size of approximately 100 kDa, similar to unactivated WT ZAK (Fig. 5d, lanes 1 and 4). Treatment with the ZAK inhibitor nilotonib during the 24-h expression led to full stabilization of ZAK(F368C) and ZAK(K387D) (Fig. 5d, lanes 6 and 7). Treatment with MLN4924 to inhibit cullin-RING-mediated protein degradation[45] similarly stabilized ZAK levels during the 24-h expression period, allowing us to visualize the highly phosphorylated form of ZAK(F368C) and ZAK(K387D) in untreated cells (Fig. 5d, lanes 10 and 11). As expected, ANS treatment induced autophosphorylation of WT ZAK but did not change the extent of phosphorylation of F368C and K387D, which were already fully activated before ANS treatment (Fig. 5d, lanes 13–15). Finally, ANS treatment did not cause any change in the phosphorylation of K394D, supporting the hypothesis that this mutation abrogates ZAK kinase activity, probably by interfering with the SAM–SAM interface (Fig. 5d, lane 16).

Together, these data support a model in which the SAM domain has a central role in ZAK activity. We hypothesize that the ZAK SAM domains need to form a head-to-tail dimer on the colliding ribosomes that results in kinase activation and/or relief of kinase autoinhibition. We explored this hypothesis through cryo-EM analysis of a native pull-down of ZAK(K45M) harbouring the inactivating K394D mutation (Extended Data Fig. 8). The 3D reconstructions revealed ZAK bound to both stalled and collided ribosomes via the pin and RIH including the RIH-p (Extended Data Fig. 4g), as for all previous structures; however, we found that the RIM connected to its SAM domain only on RACK1(c) and no SAM dimer bridging stalled and collided ribosomes (Fig. 5e and Extended Data Fig. 4g). On the stalled ribosome, instead of the ZAK–RIM interaction, we observed SERBP1 bound to RACK1 and the second SAM domain completely missing.

## Kinase regulation is ribosome independent

To ask whether hyperactive mutants can indeed bypass collision-dependent activation, we combined RIH and RIM mutations (which disrupt ZAK binding and activation; Figs. 3f and 4f) with the hyperactive F368C mutation. These compound mutants with F368C revealed constitutive activation of ZAK (independent of ANS treatment; Fig. 5f, lanes 2–5) and thus independent of collisions or any ribosomal interaction. These data suggest that WT ZAK relies on specific ribosome interactions to relieve autoinhibition and to promote kinase activation, but that this mechanism can be circumvented by directly altering the SAM domain. The crystal structure of ZAK kinase domains complexed with vemurafenib is a dimer consistent with the idea that proximity of the kinase domains is important[27]. AlphaFold predicts various alternative conformations of the SAM dimer (Extended Data Fig. 4i), but the asymmetric head-to-tail interface dimer observed in the disome structure was only predicted for WT SAM domains and not for any of the described mutants. We speculate that the SAM domains have a role in stabilizing the inactive state of ZAK and that structural rearrangements caused either by (pathogenic) mutations in the SAM domain or facilitated by dimer formation after binding to the disome promote activation[46].

## Discussion

Our structural and biochemical data reveal that the ZAK-driven RSR is activated by the unique interface of a collided disome. Although we cannot exclude the possibility of individual stalled ribosomes being sufficient for ZAK activation, our data strongly support a model in which collisions have a key role. Through specific interactions between ZAK and ribosomal proteins (RACK1 and eS27) as well as rRNA (ES6c and ES7), the SAM domains of ZAK are oriented to dimerize and promote kinase activation. These data reveal how the ribosome acts as a scaffold, initially for global recruitment of ZAK and then for ZAK kinase activation upon collision. In the case of disome

formation after translational stalling, RACK1 reveals itself as the key scaffold and the central player for collision sensing and ZAK activation. These observations are reminiscent of the critical role of RACK1 in ribosome-associated quality control[20,47,48], although the structural specifics for recruitment and activation of the E3 ligase ZNF598 (Hel2 in yeast) remain unknown.

Previous biochemical and sequencing data have revealed the presence of ribosome collisions even under untreated basal conditions[49]. Multiple modes of regulation probably work together to prevent premature RSR signalling during unstressed conditions. First, the stoichiometry of ZAK to ribosomes (1:100) ensures that ZAK can only be bound to a small fraction of the ribosomes at a given time. This suggests a sampling model (Fig. 5g) in which ZAK (as a monomer or dimer) transiently binds to monosomes and disomes. We postulate that ZAK more stably binds to disomes due to a higher local concentration of binding sites and due to additional binding contacts only available at the disome interface, eventually leading to kinase activation. The RIH-p is one example of a region of ZAK that may provide this specificity by bridging the stalled and collided ribosome. Another example is the ES6-patch, specifically bridging to the collided ribosome (Fig. 2c and Extended Data Fig. 6b). In addition, under basal conditions, disomes may not be long lived or abundant and ribosome-associated quality control factors can resolve collisions before ZAK has sufficient time for activation.

Another moderating influence is that the ZAK RIM, which binds to stalled and collided ribosomes, must compete with the abundant factor SERBP1 at the FPxL-binding site of RACK1; this competition probably serves as a source of basal negative regulation (Fig. 4i). We hypothesize that this FPxL motif (present in ZAK and SERBP1) may have broad implications for other ribosome-associated quality control factors that may also rely on this RACK1 interaction, with LARP4 being one example[50] (Extended Data Fig. 9).

It is not clear from our structures how the reorientation and stable dimer formation of the SAM domains on collided ribosomes is communicated to the other regions of ZAK for activation and catalysis. We hypothesize that SAM dimer formation on the stalled and collided RACK1 proteins facilitates a conformational rearrangement, possibly communicated through the leucine zippers and/or the YLD, that promotes ZAK activation by an allosteric mechanism. We identified hyperactivating SAM dimer interface mutants that promote kinase activation (Fig. 5d) and might prove useful to understand the molecular principles of kinase activation in future studies. Together, this work provides mechanistic insights into how ZAK, a central switch for cell-fate regulation, senses stress directly on the translational machinery upon collisions and proposes a conclusive model for how this leads to ZAK kinase activation.

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

# Methods

## Generation of the knockout cell line (*ZAK* and *RACK1* knockout)

HEK293T *RACK1*-knockout cells were generated by CRISPR–Cas9 (refs. 51,52). The single guide RNA (sgRNA; *ZAK* target sequence: TGTATGGTTATGGAACCGAG; *RACK1* target sequence ACTGCGGGGT AGTAGCGATCTGG) was subcloned into the pX330 plasmid[51]. HEK293T cells (American Type Culture Collection CRL-3216) were transfected with 1.5 μg of plasmid using Lipofectamine 3000 (L3000075, Thermo) according to the manufacturer's instructions. After 2 days, cells were collected and plated on 96-well plates (3603, Corning) by limiting dilution. Colonies were confirmed for *ZAK* or *RACK1* deletion by immunoblotting and sequencing.

## Tissue culture

HEK293T cells were maintained using Dulbecco's modified eagle medium (DMEM; 11995073, Thermo) supplemented with 10% FBS (A3160502, Thermo Fisher) and passed using trypsin-EDTA (0.25%) and phenol red (25200114, Thermo). Cells were seeded at $2 \times 10^6$ (15 cm; CLS430599, Millipore Sigma), $1.5 \times 10^6$ (10 cm; CLS430167, Corning) or $3 \times 10^5$ cells per well (six well; 3516, Fisher). At 24 h, cells were transfected using Lipofectamine 3000 Transfection Reagent (L3000075, Thermo) according to the manufacturer's instructions. Twenty-four hours post-transfection, cells were treated and lysed (approximately 70% confluency). Medium was changed 1–2 h before drug treatment and/or lysis. ANS (A9789, Sigma) was added directly to the media (untreated = DMSO (D12345, Thermo); ANS collision dose = 0.38 μM final concentration). ANS stock solutions were 94.2 mM (25 mg ml$^{-1}$) in DMSO. Unless noted, all ANS treatments were done for 15 min. For the protein stabilization experiments, medium was supplemented with 2 μM MLN4924 (B1036, ApexBio) or 2 μM nilotonib (A8232, ApexBio) from the time of transfection to time of harvesting (24 h). To end any treatment, medium was aspirated and cells were quickly washed with ice-cold PBS (10010-049, Thermo; 8 ml for 10 cm; 2 ml for six well) and then the lysis buffer (50 mM HEPES pH 7.5, 100 mM KOAc, 5% glycerol (G33-4, Fisher), 0.5% Triton X-100 (T9284, Millipore Sigma), 15 mM Mg(OAc)$_2$, 1× Halt protease + phosphatase inhibitor cocktail (78445, Thermo Fisher) and Turbo DNase I (80 units; AM2239, Thermo)) was directly added to plates and the cells were collected by scraping. Plates (10 cm) were lysed in 200 μl lysis buffer; six-well dishes were lysed in 100 μl lysis buffer. Lysates were kept on ice and clarified at 8,500*g* for 5 min. Clarified lysates were flash frozen in N$_2$ and stored at −80 °C.

## RNA knockdowns

Cells were seeded at $1 \times 10^6$ cells (10 cm) and $7.5 \times 10^4$ (six well). After 24 h, cells were treated with siRNA (50 μM stocks, 50 nM final concentration) using Lipofectamine RNAiMAX Transfection Reagent (13778150, Thermo) according to the manufacturer's instructions. After 24 h, the medium was changed. Seventy-two hours post-siRNA transfection, cells were treated and lysed according to the above lysis protocol.

## Sucrose gradients

Preparative (12 ml) sucrose gradients were made using 10× gradient buffer (250 mM HEPES pH 7.5, 1 M KOAc and 50 mM Mg(OAc)$_2$) to make final gradients with sucrose buffer (1× gradient buffer, 10% or 50% sucrose (60% sucrose stock), 1 mM TCEP (TCEP25, Gold-Bio) and SuperaseIN (200 units)). Approximately 25–50 μg RNA (quantified by qubit HS) was loaded on gradients in 200–300 μl final volume. The Beckman Coulter Ultracentrifuge and Beckman SW41 swinging bucket rotor were used for centrifugation. For regular gradients (10–50% sucrose), spins were done at 274,000*g* for 1 h 45 min at 4 °C. Ten fractions were collected and absorbance at 260 nm (A$_{260}$) was measured using the Biocomp Piston Gradient Fractionator. Trichloroacetic acid (TCA; T3699, Millipore Sigma) was added to each fraction (10% final

concentration). Samples were frozen at −20 °C overnight. The TCA precipitation protocol followed.

Analytical (200 μl) gradients were made by stacking 40 μl of 50%, 40%, 30%, 20% and 10% sucrose buffer in 250 μl tubes (343775, Beckman). Approximately 1–2 μg RNA (quantified by qubit or normalized by bicinchoninic acid (BCA)) was loaded in 10 μl final volume on a 10–50% 200 μl gradient. The Beckman Coulter Tabletop Centrifuge (CTZ24D006, Optima MAX) and TLS55 rotor were used for centrifugation. Spins were done at 214,000*g* for 22 min at 4 °C. Ten 20 μl fractions were taken and added directly to 7 μl of 4× loading buffer. Of each fraction, 8 μl was run on 4–20% TGX 26-well gel (5671095, Bio-Rad).

## Immunoblotting

Concentration-normalized samples were generated using total protein quantification (BCA assay; 23225, Thermo Fisher) and were then diluted in 4× loading buffer (8% sodium dodecyl sulfate (SDS), 40% glycerol, 0.4 mM bromophenol blue and 40 mM Tris-Cl pH 6.8) and boiled at 95 °C for 10 min. Approximately 5 μg of protein was loaded into 4–20% Criterion TGX polyacrylamide gels and run in 1× Tris-glycine running buffer (1610732, Bio-Rad) at 150 V for 1 h. Proteins were transferred to polyvinylidene fluoride (PVDF) membranes (Trans-Blot Turbo RTA Midi 0.2 μm PVDF Transfer Kit 1704273) for 10 min at 2.5 A. Membranes were blocked with 5% non-fat milk (blocking buffer; sc-2325, Santa Cruz Biotech) in Tris-buffered saline with Tween 20 (TBS-T) for 1 h at room temperature. All primary antibody incubations were done overnight at 4 °C in blocking buffer. After three 10-min TBS-T washes, secondary antibodies diluted in blocking buffer were incubated for 1 h at room temperature. Three 10-min washes with TBS-T followed secondary incubation. All incubations and washes were performed with gentle rocking. Blots were visualized using SuperSignal West Pico PLUS (34580×4, Thermo) and/or West Femto Maximum (34095, Thermo) chemiluminescent substrates and the Bio-Rad ChemiDoc imaging system.

## Phos-tag SDS–PAGE

Concentration-normalized samples were diluted in 4× loading buffer and boiled as described for immunoblotting and then loaded on 8% SDS–PAGE (10.7 μM Phos-tag acrylamide (AAL-107S1, Wako) and 21.3 μM MnCl$_2$). Samples were run in 1× Tris, glycine and SDS running buffer (125 V for 2.5 h). An EDTA-free pre-stained protein marker (F4005, Apex Bio) was used. Two 10-min washes in 1× transfer buffer (25 mM Tris, 192 mM glycine and 10% v/v methanol) supplemented with 1 mM EDTA (AM9260G, Thermo) were followed by two 10-min washes in 1× transfer buffer without EDTA. Samples were transferred to PVDF membranes (1620177, Bio-Rad) overnight at 35 V (room temperature). Blocking and antibody incubation and visualization are described in the section 'Immunoblotting'.

## TCA precipitations

Sucrose fractions in 10% TCA were thawed on ice and centrifuged at 21,000*g* at 4 °C for 30 min. The sucrose and TCA were aspirated off and the remaining pellet was washed with 500 μl acetone (A18P-4, Thermo) and spun at 21,000*g* at 4 °C for 10 min. The wash step was repeated and then the pellets were dried using a vacuum evaporator for 5 min. The dried pellets were resuspended in 4× loading buffer (40–60 μl total; 10 μl loaded on gel).

## Co-immunoprecipitation of ZAK dimers

*ZAK*-knockout HEK293T cells were transfected with either FLAG-tagged or haemagglutinin-tagged ZAK constructs (24-h expression) and then treated and lysed from six-well plates (see the section 'Tissue culture'; lysis buffer: 50 mM HEPES pH 7.5, 100 mM KOAc, 5% glycerol, 1% digitonin (D-180-1, Gold Bio), 5 mM Mg(OAc)$_2$, Halt protease + phosphatase inhibitor cocktail (100×) and Turbo DNase I (80 units)) according to the above tissue culture protocols (six well, 100 μl lysis buffer). After scraping, the cells were incubated on ice for 30 min before clarification

and flash freezing. On the day of the experiment, cells were thawed on ice and normalized using $A_{260}$ as measured by nanodrop (13-400-529, Thermo Scientific). MNase buffer (0.2×; B0247S, NEB) with 1 mM $CaCl_2$ and 500 U MNase (M0247S, NEB) were added to normalized lysates and the reaction was incubated at 22 °C for 30 min. The samples were moved to ice and 2 mM EGTA (50-997-744, Fisher) was added. For the co-immunoprecipitations, 90 µl of sample was added to 2 µl washed anti-FLAG M2 magnetic affinity resin (M8823, Millipore) and incubated for 90 min with rotation at 4 °C. The samples were washed three times with 100 µl wash buffer (50 mM HEPES pH 7.5, 100 mM KOAc, 5% glycerol, 0.1% digitonin, 5 mM $Mg(OAc)_2$ and Halt protease + phosphatase inhibitor (100×)) for 5-min incubation on nutator at 4 °C. The protein was eluted (elution buffer: wash buffer plus 400 µg ml$^{-1}$ 3×FLAG peptide) with two 10 µl elutions (30 min incubation on rocker at 4 °C). Samples were added to 4× loading buffer and immunoblots were performed according to the above described methods.

## CLIP sample preparation

CLIP-seq samples were prepared as previously described[53]. Two 10-cm plates per construct of HEK293T ZAK-knockout cells (pcDNA3.1-FLAG-ZAK_T161/S165A, pFN24K_3×FLAG-ZAK, pFN24K_3×FLAG-ZAK_T161/S165A and pFN24K_3×FLAG-ZAK_1-649) were transfected in addition to two 10-cm mock no DNA control plates. After 24 h of expression and 15 min of ANS treatment, cells were crosslinked at 254 nM UV, lysed, clarified and RNAseI digested (AM2295, Invitrogen). Immunoprecipitation was performed overnight at 4 °C using 13 µl of anti-FLAG M2 affinity resin (A2220, Millipore). Following immunoprecipitation, samples were washed, dephosphorylated with FastAP enzyme (EF0654, Thermo) and T4 PNK (M0201L, NEB) and the 3′ RNA adapter was ligated (M0437M, NEB). A small amount of sample was run on a Criterion XT 4–12% Bis-Tris gel and transferred to PVDF to perform a diagnostic western blot probing for ZAK expression and size. The remaining sample was run on a Criterion XT 4–12% Bis-Tris gel (3450124, Bio-Rad) and transferred to a nitrocellulose membrane and a region was cut corresponding to the ZAK protein size to plus approximately 75 kDa. Membrane pieces were digested with proteinase K (P8107S, NEB). RNA was purified on a clean and concentrator column and reverse transcribed with SuperScript III (18080044, Thermo). The RNA and free primer were digested and the RT-DNA purified with MyOne Silane beads (37002D, Thermo) and the 5′ DNA adapter was ligated (M0437M, NEB). Following cleanup, a pilot quantitative PCR was performed. Samples were amplified for the determined number of PCR cycles, gel extracted for products corresponding to 175–35 bp and submitted for next-generation paired-end sequencing with 2 × 150-bp read length.

## CLIP analysis

All code for CLIP-seq analysis has been published on GitHub (https://github.com/jakesaba/2025_ZAK). In brief, unique molecular identifiers were appended to each paired-end read using umi_tools extract[54] and trimmed using trim_galore (https://github.com/FelixKrueger/TrimGalore). Reads were aligned using STAR[55] to the GRCH38 genome containing a single ribosomal DNA (chrR), originally generated by the Paralkar laboratory[56]. Aligned reads were sorted and indexed using samtools[57] and deduplicated using umi_tools dedup.

For mapping coverage to 18S rRNA, bam files were imported into R, coverage was normalized to library size and then mean-scale normalized across the 18S region. Mean-scaled coverage over 18S was then normalized to the coverage of the ZAK_1–649 truncation[1] at each position. To avoid dividing by 0, a pseudocount corresponding to the 0.1 percentile signal was applied to the coverage of the ZAK_1–649 sample at all positions. To reduce noise, nucleotide positions corresponding to less than 3% of the cumulative CLIP-seq coverage signal were removed and their fold enrichment was set equal to 1. Plots were smoothed using a rolling average with a window size of 10.

For genome-wide analysis of CLIP-seq peaks, a similar approach was used with a few exceptions. First, no mean-scale normalization was applied and coverage was normalized to the ZAK-knockout sample. A global pseudocount of 5 was applied and cumulative signal less than 3% of the cumulative CLIP-seq coverage at each gene locus was again removed. CLIP peaks with average reads per million of more than 10 and satisfying a more than twofold enrichment over a window size of more than 20 compared with the ZAK knockout were called. Significance was determined using a one-sided Poisson test. For significant peaks, a false discovery rate was assigned using the Benjamini–Hochberg procedure. For each gene, only the canonical transcript was used.

For metagene analysis, we aligned CLIP-seq coverage data to standardized transcript regions (5′ untranslated region (UTR), the coding sequence (CDS) and 3′ UTR). For each gene, only the canonical transcript was used, and only transcripts with a CDS length of at least 300 nucleotides were retained. For each transcript, the 5′ UTR, CDS and 3′ UTR were separately scaled to 100 positions, and coverage values were linearly interpolated to create a fixed-length alignment across all genes. These were concatenated to produce a 'metagene' axis of 300 standardized positions (0–100 for 5′ UTR, 100–200 for CDS and 200–300 for 3′ UTR). To account for background signal, the metagene profile of each condition was normalized to a ZAK-knockout control profile, computed as the ratio of the mean signal to the ZAK-knockout signal at each metagene position. For visualization, smoothed profiles (rolling average, window size = 5) were plotted with region boundaries clearly marked.

## Bacterial expression and purification of ZAK RBR

For ZAK RBR (C terminus 100 amino acids) protein expression, the sequence was cloned into pGEX backbone with a N-terminal GST tag. The plasmid was transformed into BL21-competent cells (C2527I, NEB) and allowed to outgrow overnight at 37 °C. The starter culture was added to a 1-l flask of 2× YT media (31GE58, Grainger), and at optical density at 600 nm of 0.6, protein expression was induced with 1 mM IPTG for 2 h. Bacterial pellets were collected (4,000g for 10 min), flash frozen and stored at −80 °C.

Pellets were thawed on ice in lysis buffer (50 mM Tris pH 8, 150 mM NaCl, 5% glycerol, 1 mM TCEP, 0.2 mM phenylmethylsulfonyl fluoride (PMSF; P7626-25G, Sigma), 1× EDTA-free cOmplete protease inhibitor tablet (5056489001, Sigma), pinch of DNase I (10104159001, Millipore Sigma) and pinch of lysozyme (L6876, Sigma)) to a final volume of 50 ml and dounced on ice until fully resuspended. The lysate was sonicated at 50% amplitude (3 s on; 10 s off; 1 min total) before clarification using the TI45 rotor and spinning at 186,000g for 30 min. After the spin, the supernatant was filtered using 0.45-µM filter (431220, Corning) and loaded onto GSTrap 5 ml column (17513102, Cytiva) using the Cytiva (GE Healthcare) AKTA Pure FPLC system. After binding, the column was washed with wash buffer 1 (50 mM Tris pH 8, 150 mM NaCl, 1 mM TCEP, 0.2 mM PMSF and 1 protease inhibitor pill) and high-salt wash buffer 2 (50 mM Tris pH 8 and 1 M NaCl). The protein was eluted (elution buffer: 50 mM Tris pH 8, 300 mM NaCl and 10 mM reduced glutathione) and the eluted fractions were pooled and concentrated using Pierce Protein Concentrators PES, 30 K MWCO (88522, Thermo) to 1 ml before size-exclusion chromatography (Cytiva Superdex 75) with SEC buffer (50 mM HEPES pH 7.5, 300 mM KOAc, 5 mM $Mg(OAc)_2$, 5% glycerol and 1 mM TCEP). Protein samples were concentrated, flash frozen and stored at −80 °C until use.

## Native pull-downs of ZAK-bound ribosomal complexes for cryo-EM

Expi293F cells (A14527, Thermo Fisher) transiently transfected with the pcDNA3.1-FLAG-ZAK-K45M construct were treated with ANS (0.38 µM) for 15 min and collected in lysis buffer (50 mM HEPES pH 7.5, 150 mM KOAc, 5 mM $Mg(OAc)_2$, 1 mM dithiothreitol, 0.5% NP-40 and EDTA-free protease inhibitor cocktail (Roche)). Cells were

homogenized using a dounce homogenizer (DWK Life Science) and clarified by centrifugation at 36,603$g$ for 15 min at 4 °C. The supernatant was treated with Nuclease S7 (20 U ml$^{-1}$; Sigma-Aldrich) for 15 min at 25 °C. Digested lysates were incubated with Anti-FLAG M2 agarose beads (Sigma-Aldrich) on a rotating wheel for 3 h at 4 °C. Beads were transferred to a 1-ml Mobicol column (MoBiTec) and washed twice with 10 ml of wash buffer (50 mM HEPES pH 7.5, 150 mM KOAc, 5 mM Mg(OAc)$_2$, 1 mM dithiothreitol and 0.01% NP-40). Complexes were eluted in elution buffer (20 mM HEPES pH 7.5, 150 mM KOAc, 5 mM Mg(OAc)$_2$, 1 mM dithiothreitol, 0.05% Nikkol and 300 ng μl$^{-1}$ FLAG peptide (Sigma-Aldrich)) for 1 h at 4 °C.

The same purification protocol as described above was also used for the Strep–ZAK(T161A/165A) and FLAG–ZAK(K45M/K394D) pull-downs as well as for the FLAG–ZAK(K45M) pull-down performed without previous challenging cells with ANS (see also Extended Data Fig. 1b).

### In vitro binding assays and reconstitutions of ZAK RBR–ribosome complexes

Human ribosomal subunits and 80S monosomes were purified from Expi293F cells. Cells were lysed in lysis buffer (50 mM HEPES (pH 7.5), 150 mM KOAc, 5 mM Mg(OAc)$_2$, 0.5% NP-40, 1 mM dithiothreitol, 1 mM PMSF and EDTA-free protease inhibitors). Lysates were clarified by centrifugation at 36,603$g$ for 15 min, loaded onto 10–50% sucrose gradients and spun at 284,600$g$ for 3.5 h at 4 °C using a SW40Ti rotor (Beckman Coulter). Gradients were fractionated into 500-μl fractions to separate 40S and 60S ribosomal subunits from 80S monosomes. 40S, 60S and 80S fractions were pooled and pelleted through a sucrose cushion using a TLA110 rotor (Beckman Coulter) at 460,800$g$ for 45 min at 4 °C, then resuspended in binding buffer (50 mM HEPES pH 7.5, 150 mM KOAc, 5 mM Mg(OAc)$_2$, 1 mM dithiothreitol and 0.01% NP-40).

For the in vitro binding assay, the GST-3C-tagged ZAK RBR protein (see above) was incubated with either purified ribosomal subunits or monosomes (25 pmol each) for 60 min at 4 °C. Reactions were diluted with 360 μl binding buffer and transferred to 1-ml Mobicol columns (MoBiTec) containing 20 μl glutathione Sepharose 4 fast flow resin (Cytiva) and incubated for 60 min at 4 °C. Beads were washed three times with binding buffer (1 × 800 μl, 2 × 500 μl). Bound complexes were eluted with binding buffer containing 25 mM reduced L-glutathione (Sigma-Aldrich) for 60 min at 4 °C. Samples were analysed on a 12% polyacrylamide gel (Invitrogen) and stained with Der Blaue Jonas (German Research Products).

### Electron microscopy and image processing

For all cryo-EM samples, grids were prepared and images were processed the same way. All samples were crosslinked with 0.02% (v/v) glutaraldehyde on ice for 20 min. Reactions were quenched by addition of Tris-OAc to a final concentration of 25 mM. Of each sample, 3.5 μl (approximately 4–8 A$_{260}$ per ml) was applied to Quantifoil R3/3 holey carbon grids with 2-nm continuous carbon coating, blotted for 3 s and then plunge frozen in liquid ethane using a Vitrobot Mark IV (Thermo Fisher Scientific). Data collections were performed at 300 keV using a Titan Krios microscope equipped with a Falcon 4i direct electron detector and a SelectrisX imaging filter using EPU software (3.7; all Thermo Fisher Scientific) at a pixel size of 0.727 Å. Dose-fractioned movies were collected in a defocus range from −0.5 to −3.5 μm and with a total dose of 40 e$^-$ Å$^{-2}$, fractionated in 40 frames to obtain a total dose of 1 e$^-$ Å$^{-2}$ per frame. Gain correction, movie alignment and summation of movie frames were performed using MotionCor2 (v1.4.0)[58]. Contrast transfer function parameters were estimated using CTFFIND4 (v4.1.13)[59].

Structures of ZAK–disome complexes were obtained from native pull-downs after treatment with ANS using either the FLAG-tagged ZAK(K45M) mutant or the ZAK(T161A/165A) mutant.

From the combined datasets, 2,246,220 particles were automatically picked from a total of 95,813 micrographs in RELION (v5.0 beta)[60].

After 2D classification in CryoSPARC (v4.6.0)[61], 1,347,732 80S ribosomal particles were selected and 3D classified in RELION. This yielded 80S classes with strong density for neighbours, indicative of stable disomes, or with no or only weak extra density for neighbours. 80S with no neighbour density represented either 80S with mRNA and tRNAs in hybrid state (A/P and P/E) or POST state (P/P and E/E), or 80S bound to eEF2 (and SERBP1). Disome classes (80S classes with strong neighbour density) occurred either with a neighbour at the mRNA exit site or at the mRNA entry site, defining them as stalled and collided 80S, respectively. Both stalled and collided 80S were found in POST and in hybrid states, and each of those disome classes displayed additional density at RACK1, accounting for the ZAK SAM dimer. The two most abundant classes (POST-state stalled 80S and hybrid-state collided 80S) were further processed. They were classified with a soft mask focusing on the RACK1 region where extra density for ZAK was found, revealing subclasses with the ZAK SAM dimer varying in flexibility. Classes displaying strong SAM density were joined and refined to an overall resolution of 2.3 Å for both the stalled 80S and the collided 80S, and then subjected to local refinement in CryoSPARC. Local resolution was determined for the RACK1 SAM regions to be between below 3 Å (for the RACK1–RIM interaction) and between 5 Å and 8 Å for the globular SAM domain. Finally, for a sub-dataset, two maps for the entire ZAK–disome complex were generated by extending the box size of both stalled and collided 80S, respectively, and centring the disome density. Both maps showed low-resolution extra density adjacent to the SAM dimer, probably accounting for the leucine zipper–kinase domain (LZ–KD) dimer. The disome obtained from extending the stalled 80S was locally refined focusing on the RACK1 SAM region, yielding a clear density for the SAM dimer at a local resolution between 6 Å and 11 Å and served as consensus disome map. The disome obtained from extending the colliding 80S was used for fitting the LZ–KD dimer.

The composite map for the ZAK–disome was assembled by first fitting the individually refined stalled and collided 80S and the locally refined maps into the 6 Å resolution disome consensus map and then using the 'vop max' tool in UCSF Chimera X (v1.9)[62] to join the individual maps (Extended Data Fig. 2).

The sample obtained from the native FLAG–ZAK(K45M) pull-down without ANS treatment was processed as described above. Here, however, 3D classification of 325,370 particles picked from a total of 13,154 micrographs showed no classes indicative for stable disomes. Instead, the main classes represented two classes of translating 80S, one with tRNAs in hybrid state (A/P and P/E) and one with tRNAs in POST state (P/P and E/E), as well as 80S with a tRNA in the E site and bound to eEF2 and SERBP1, indicative of hibernating ribosomes[38,39]. The hybrid state translating and the hibernating 80S classes were first refined followed by local refinement focusing on either the entire 40S subunit (2.3 Å for the translating and 2.7 Å for the hibernating 40S) or the RACK1 region (2.7 Å for the translating and 3.0 Å for the hibernating 40S head; Extended Data Fig. 3a).

For the sample obtained from the FLAG–ZAK(K45M/K394D) pull-down, 3D classification of 629,553 particles picked from a total of 56,746 micrographs yielded similar 80S classes as described for the pull-downs using kinase-inactive ZAK mutants described above. Among them were classes representing disomes as well as hibernating (with eEF2/SERBP1 and E tRNA) and translating (hybrid and POST state) 80S. The classes representing stalled (with P/P and E/E tRNAs) and collided (with A/P and P/E tRNAs) 80S were refined followed by local refinement focusing on the RACK1 SAM region. Local resolution was determined for this region to be below 3 Å close to RACK1 and to between 5 Å and 15 Å for peripheral regions. We observed density for one SAM globular domain emerging from RACK1 of the collided ribosome, whereas on RACK1 of the stalled one is occupied by SERBP1 and the ZAK–RIH (Extended Data Fig. 8).

For the reconstituted ZAK–RBR–40S sample (obtained from the in vitro binding assay), 873,389 particles were picked from a total of 19,851 micrographs. 3D classification yielded 629,046 particles of 40S

that were further refined, followed by local refinement focusing on either the head (2.1 Å) or the eS27 (2.3 Å) region (Extended Data Fig. 5d).

## Model building and refinement

The disome model was generated by rigid-body fitting known 80S monosome structures into the cryo-EM density. For the stalled 80S, the ribosome structure (Protein Data Bank (PDB) ID 8GLP)[30] representing a POST-state 80S with mRNA and P-site tRNA, bound ANS was used; for the collided 80S ribosome, the structure of the human ribosome in the hybrid PRE state (PDB ID 6Y57)[29] served as a template. E-site tRNA model from the human disome (PDB ID 7QVP)[34] was used and fitted into density on the stalled 80S map. The tips of ES6c (690–740) and ES6b (741–800) were built based on AlphaFold3 prediction of 18S rRNA fitting into low-pass-filtered density on the collided 80S and the stable disome reconstructions (Extended Data Fig. 2). The mRNA model on the collided 80S from PDB ID 6Y57 was changed from 46-UUU-48 to AUG and 49-UUU-51 to UUC.

For the EDF1 C-terminal part (residues 24–145) on the collided 80S, the existing model from (PDB 6ZVH)[21] was fitted. For the N-terminal part (residues 7–14) on the stalled 80S, the AlphaFold3 prediction for the interaction of EDF1 with eS26 was used for fitting (Extended Data Fig. 5f,g). A rod-like density near uS4 on the collided 80S was identified as the C terminus of eS1 (254–264) and was modelled based on AlphaFold3, predicting an interaction between eS1 residues 231–264 with uS4. These models were processed by manual real-space refinement in WinCoot (v0.9.8.93)[63] and merged into a disome model followed by real-space refinement in Phenix (v1.20.1-4487)[64].

AlphaFold2 multimer models of ZAK full length and RACK1 revealed two interacting regions: one is ZAK 611–617 (RIH) intercalating between blade 5 and blade 6 of RACK1, and the second is ZAK 417–422 (RIM) stretching across RACK1 blade 2 and blade 3. This model served as a template to match extra densities identified at the RACK1 (Extended Data Fig. 5a–e).

To adjust RIM and RIH, respectively, we then generated AlphaFold3 models of ZAK 325–425 and RACK1, and models of ZAK 600–631 and RACK1. The resulting models were fitted into the corresponding density with only minor adjustments in Coot (Extended Data Fig. 5a–e).

The main density emerging from both RACK1 proteins at the disome interface corresponds to two SAM (residues 328–416) domains extending from the N terminus of RIM, prompting us to model a SAM dimer using AlphaFold2 multimer. Among the predicted dimer models, only the one representing the asymmetric head-to-tail interface fitted our density as a rigid body. Here manual adjustment was required only for helix α5 (393–416) on both SAM domains to fit into the clearly resolved rod-like density extending from the RACK1-bound RIM peptide. An additional rod-like extra density packed against the SAM domain on the stalled 80S was interpreted as an α-helix formed by residues 568–583 of ZAK ('helix'), based on the AlphaFold database (https://alphafold.ebi.ac.uk/). The pin was identified by running AlphaFold3 predictions of the RBR region (701–800) and eS27, and then fitted into the corresponding density followed by minor adjustment in Coot.

A model for the C terminus of SERBP1 (SERBP1-C; residues 393–408) bound to RACK1 together with the ZAK RIH was generated with AlphaFold3 and adjusted using the 2.9 Å resolution map of the locally refined RACK1 from the hybrid-state translating ribosome (Extended Data Fig. 3a) followed by Phenix refinement.

AlphaFold3 models for the LZ–KD dimer (1–330) were docked into an additional globular density on the entire ZAK–disome complex when low-pass filtered to approximately 30 Å (Extended Data Fig. 2).

The ZAK model and disome model were later merged and further refined in Phenix. Model statistics were calculated using the MOLPROBITY implementation in PHENIX[65], and can be found in Extended Data Table 1.

All structural figures were prepared using UCSF Chimera X (v1.9)[62].

## Statistics and reproducibility

Unless otherwise noted, all biochemical experiments and cell-based assays were repeated a minimum of two times (in part or in whole) and the two independent replicates showed similar results.

## Antibodies used in study

The primary antibodies used were: rabbit anti-eS24 (ab196652, Abcam; 1:1,000); mouse anti-FLAG (A8592, Sigma; 1:5,000); rabbit anti-haemagglutinin (3724, Cell Signaling; 1:1,000); mouse anti-JNK1 (3708, Cell Signaling; 1:1,000; 'total JNK'); rabbit anti-phospho-SAPK/JNK (4668S, Cell Signaling; 1:1,000; 'JNK-phospho'); rabbit anti-RACK1 (5432S, Cell Signaling; 1:1,000); rabbit anti-SERBP1 (NBP1-85660, Novus; 1:1,000); mouse anti-STREP (71591-3, Sigma; 1:5,000); mouse anti-vinculin (sc-73614, Santa Cruz; 1:2,000); and rabbit anti-ZAK (A301-993A, Fortis; 1:1,000).

The secondary antibodies used were: anti-mouse (7076S, Cell Signaling; 1:5,000) and anti-rabbit (7074S, Cell Signaling; 1:5,000).

## Oligonucleotides used in study

Non-targeting (scramble) siRNA (D-001810-01-20, Horizon Dharmacon) and SERBP1-targeting siRNA (L-020528-01-0005, Horizon Dharmacon) were used.

## Reporting summary

Further information on research design is available in the Nature Portfolio Reporting Summary linked to this article.

## Data availability

CLIP-seq data have been deposited in the Gene Expression Omnibus (GSE299329). The cryo-EM structural data generated in this study have been deposited in the Electron Microscopy Data Bank and the PDB, respectively, under the following accession codes: EMD-54172 for the composite ZAK–disome (obtained from kinase inactive ZAK pull-down with ANS treatment; PDB ID 9RPV); EMD-54140 for the stalled 80S and EMD-54141 for the collided 80S (related to the composite map); EMD-54148 for the locally refined ZAK–RACK1 region of the stalled 80S and EMD-54147 for the locally refined ZAK–RACK1 region of the collided 80S (both related to the composite map); EMD-54191 for the human disome with ZAK (related to the composite map); EMD-54149 for the hybrid-state translating 80S and EMD-54150 for the hibernating 80S (obtained from native FLAG–ZAK(K45M) pull-down); EMD-54236 for the locally refined RACK1 region of the hybrid-state translating 80S (PDB ID 9RSX); EMD-54165 for the in vitro reconstituted ZAK–RBR–40S complex; EMD-54166 for the stalled 80S and EMD-54167 for the collided 80S (obtained from FLAG–ZAK(K45M/K394D) pull-down with ANS treatment). The structures used for atomic model building of ZAK-bound disome complexes are available from Worldwide PDB (wwPDB) with the accession codes 6Y57, 6ZVH, 7QVP and 8GLP.

## Code availability

All code for CLIP-seq analysis has been published on GitHub (https://github.com/jakesaba/2025_ZAK).

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

**Acknowledgements** We thank C. Ungewickell, S. Rieder, A. Gilmozzi, J. Musial and J. Brunelle for excellent technical assistance; A. Lentzch and N. Sinha for critical reading; and J. Thorner, S. Regot, L. Chitoiu, C. Hernandez Rosales and M. Thoms for valuable scientific input. Work was supported by the Howard Hughes Medical Institute to R.G., by the European Research Council (ADG 885711 Human-Ribogenesis) and the DFG (BE1814/20-1 and BE1814/22-1) to R.B., by grants from the National Key R&D Program of China (2023YFC2413204) and the National Natural Science Foundation of China (32371350) to J.C., by Graduate School of Quantitative and Molecular Biosciences Munich to S.N. and T.D., by the US National Institute of Health (NIH) BCMB 5T32GM007445 and the National Science Foundation Graduate Research Fellowship Program under grant no. (DGE2139757) to V.L.H., by NIH F30 CA260910 to J.A.S., by a Howard Hughes Medical Institute fellowship of the Damon Runyon Cancer Research Foundation (DRG-2451-21) to M.A.C., and by NIH Dermatology 5T32AR074920, a JHU Provost's Postdoctoral Fellowship Program and the Dermatology Foundation's Dermatologist Investigator Research Fellowship to E.P.

**Author contributions** V.L.H. and M.A.C. developed the cell-based and biochemistry assays. V.L.H. performed the cell-based and biochemical experiments with mentorship from M.A.C. V.L.H. and J.A.S. generated the CLIP samples. S.N. purified all ZAK-bound ribosomal complexes, generated all samples for cryo-EM and processed all the cryo-EM data. S.N. generated the molecular models with the help of J.C. and T.D. S.N., together with T.B. and R.B., analysed the structures. S.N. and T.D. performed the in vitro binding assays. M.A.C. contributed the co-immunoprecipitation experiment. J.A.S. completed the CLIP data analysis. V.L.H. and M.A.C. generated the *ZAK*-knockout cell line. E.P. generated the *RACK1*-knockout cell line and related the *RACK1*-knockout experimental data. O.B. performed the cryo-EM data collection. S.N., V.L.H., T.B., R.G. and R.B. wrote and edited the manuscript with input from all authors. V.L.H. and S.N. contributed equally to this work and reserve the right to list their names first in their resumes.

**Funding** Open access funding provided by Ludwig-Maximilians-Universität München.

**Competing interests** R.G. is a member of the scientific advisory board of Alltrna, Initial Therapeutics and Arrakis Pharmaceuticals, and consulted for Bristol Myers Squibb (Celgene) and Monta Rosa Therapeutics. All other authors declare no competing interests.

**Additional information**
**Correspondence and requests for materials** should be addressed to Rachel Green or Roland Beckmann.

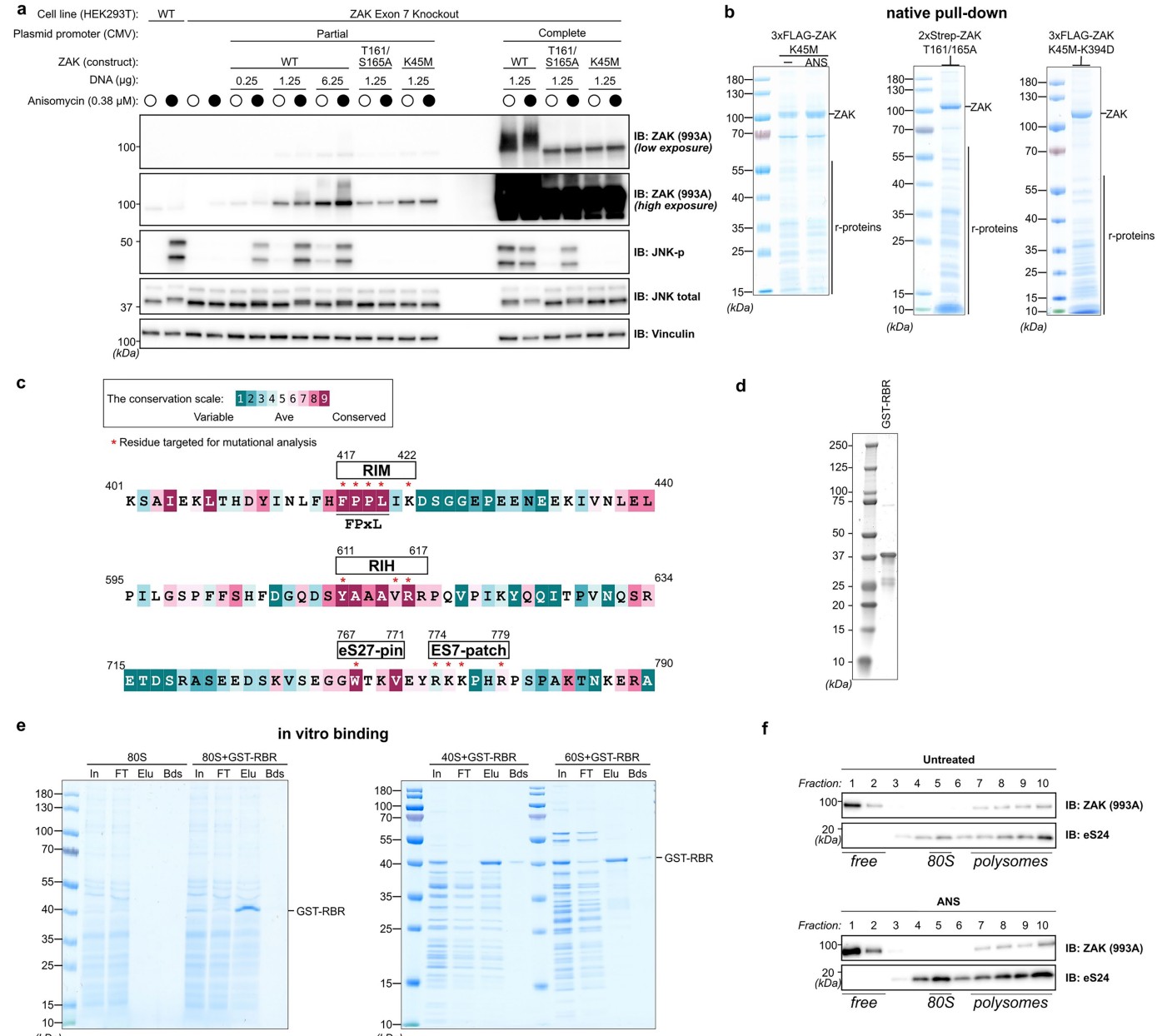

**Extended Data Fig. 1 | Conservation of ZAK and characterization of various in vivo and in vitro systems. a**, Immunoblots from wildtype (WT) HEK293Ts and ZAK knockout HEK293Ts cells without or after addition of anisomycin (−/+ ANS) and with different ZAK (mutant) constructs expressed under a partial or full CMV promoter. Plasmid DNA concentrations for transfection indicated. **b**, Coomassie stained SDS gels of elution fractions from native pull-down of tagged ZAK mutants expressed in Expi293 cells and used for cryo-EM. See Extended Data Fig. 2 for joined data from the ZAK-K45M pull-down (left panel, ANS) and the ZAK-T161/165 A pull-down (both after ANS addition), Extended Data Fig. 3a for the pull-down from untreated cells (left panel, (−)) and Extended Data Fig. 9 for the ZAK-K45M-K394D pull-down (right panel) **c**, Consurf

evolutionary conservation profile of certain ZAK regions of interest with RIM, RIH, eS27-pin and ES7-patch highlighted. Residues with red * were targeted for mutational analysis. **d**, Coomassie stained SDS gel of purified GST-tagged ribosome binding region (RBR) of ZAK. **e**, Coomassie stained SDS gels of in vitro binding assays using purified subunits (40S and 60S) or 80S ribosomes and purified GST-RBR. In = input, FT = flowthrough, Elu = elution, Bds = Beads. **f**, Immunoblots of sucrose gradient fractions collected from WT HEK293T cells. Immunoblots were probed for ZAK using 993 A antibody to visualize endogenous ZAK. Immunoblot against ribosomal protein eS24 is shown below. Blots represent at least two independent replicates (see "Statistics and Reproducibility" section). See Supplementary Figs. 8 and 9 for source data.

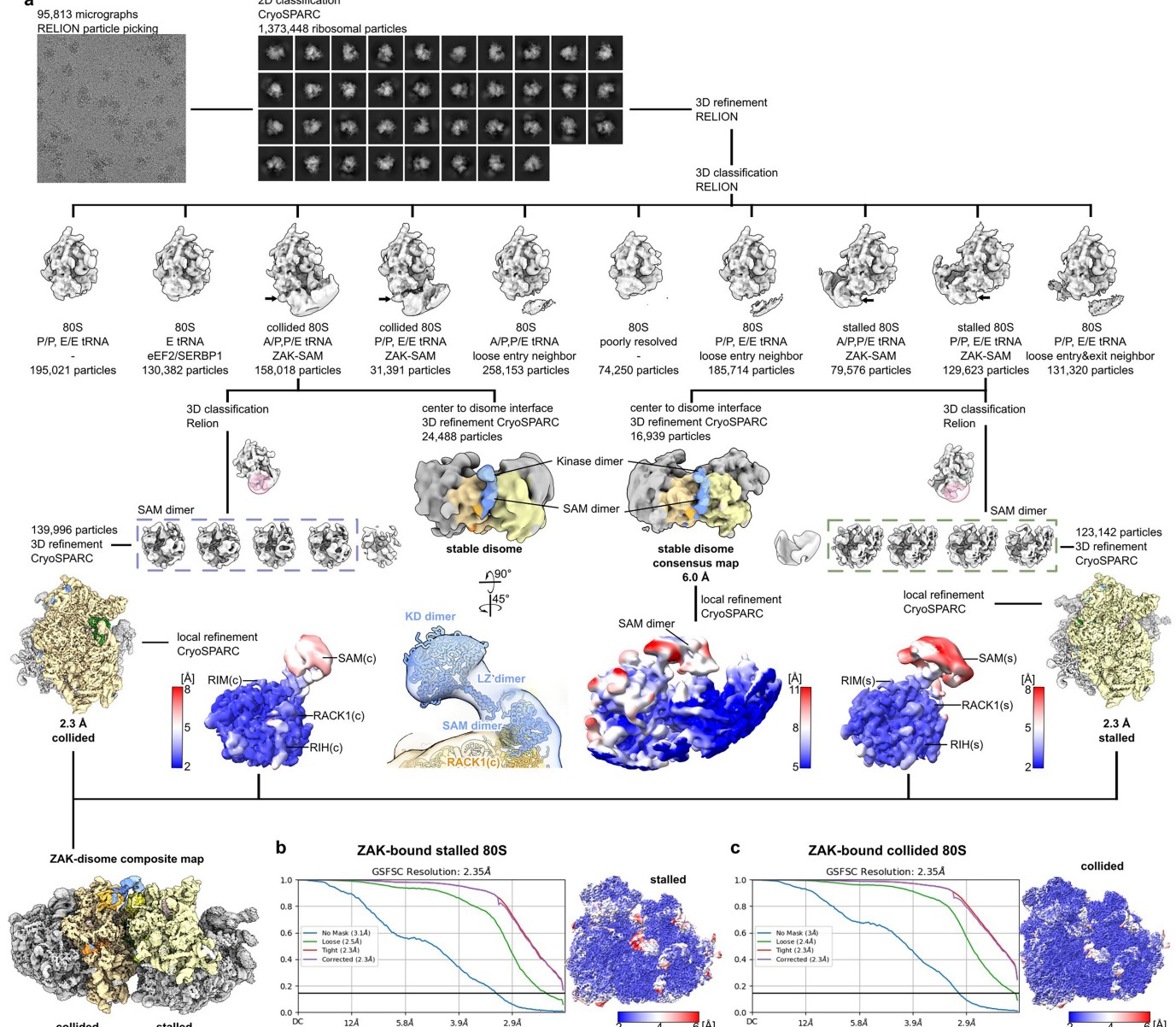

**Extended Data Fig. 2 | Cryo-EM data analysis, classification and resolution of the ZAK-disome complex.** The sample was obtained from a native pull-down using kinase inactive (K45M or T161/165 A) ZAK mutants after ANS addition (see Extended Data Fig. 1b, left and middle panels). **a**, Processing scheme: from a total of 95,813 micrographs 1,373,448 80S ribosomal particles were selected after 2D classification, 3D refined and classified. 80S classes with strong density for neighbors, indicative of stable disomes, displayed additional density at RACK1 (indicated with a black arrow), accounting for the ZAK SAM dimer. The two most abundant classes were classified with a soft mask focusing on the RACK1 region (shown in pink), revealing sub-classes with the ZAK SAM dimer varying in flexibility. Classes displaying strong SAM density were joined and refined to an overall resolution of 2.3 Å for both the stalled 80S and the collided 80S, and then subjected to local refinement. Local resolution was determined for the RACK1-SAM regions ranging from below 3 Å (for the RACK1-RIM

interaction) to 5-8 Å for the globular SAM domain, revealing the position of the ZAK RIM and ZAK RIH. From a sub-dataset, maps for the entire ZAK-disome complex were generated for both stalled and collided 80S, respectively. They are shown low-pass filtered to highlight extra density for the ZAK SAM and kinase domains (see panel with fitted AF model below). The disome obtained from extending the stalled 80S was refined to 6 Å average resolution and was used as consensus map for assembling the composite map. This was created by fitting individually refined stalled and collided 80S and the locally refined maps into the consensus disome map. Further, this disome map was locally refined focusing on the RACK1-SAM region yielding a clear density for the SAM dimer at a local resolution between 6–11 Å (see Extended Data Fig. 4e). **b, c**, Gold-standard Fourier Shell Correlation (GSFSC) curves (obtained from CryoSPARC) and cryo-EM maps of the stalled (**b**) and collided (**c**) colored according to local resolution.

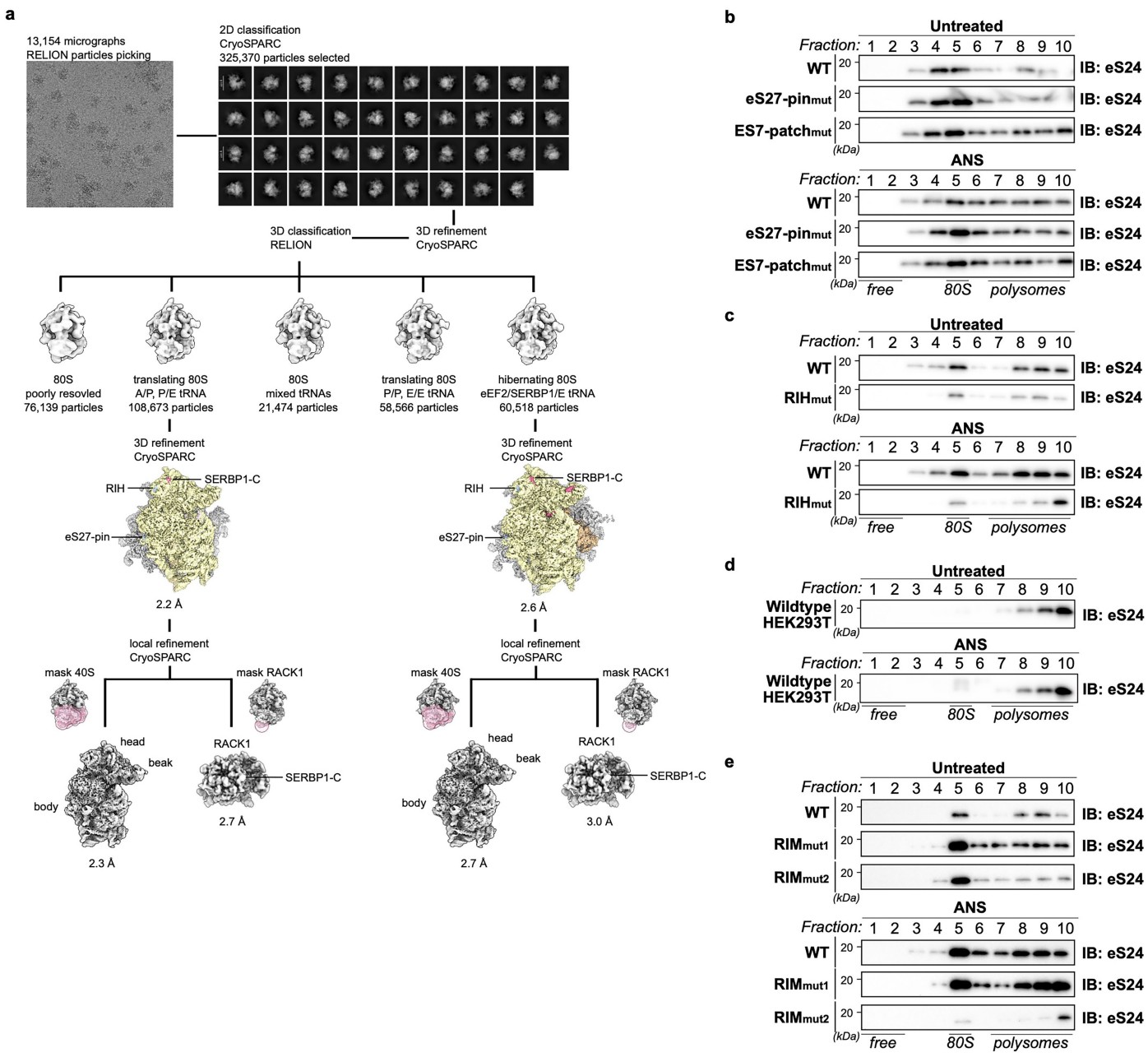

**Extended Data Fig. 3 | Cryo-EM data analysis, classification and resolution of the ZAK-bound ribosomal complexes obtained without ANS addition and eS24 control blots. a**, The sample was obtained from a native pull-down using the kinase inactive (K45M) ZAK mutant without ANS addition (see Extended Data Fig. 1b, left panel). 3D classification of 325,370 particles picked from a total of 13,154 micrographs showed 80S classes representing translating 80S with tRNAs in hybrid state (A/P, P/E) or with tRNAs in POST state (P/P, E/E) state, as well as 80S with a tRNA in the E site and bound to eEF2 and SERBP1, indicative of hibernating ribosomes. No classes indicative for stable disomes were found. The hybrid state translating and the hibernating 80S classes were first globally refined to 2.2 Å and 2.6 Å, respectively, followed by local refinement focusing on either the entire 40S subunit (2.3 Å for the translating and 2.7 Å for the hibernating 40S) or the RACK1 region (2.7 Å for the translating and 3.0 Å for the hibernating 40S head). Both maps revealed the ZAK eS27-pin, the ZAK RIH bound to RACK1 and the RACK1-bound SERBP1 C-terminal region. **b**, eS24 blots corresponding to Fig. 2e. **c**, eS24 blots corresponding to Fig. 3e. **d**, eS24 blots corresponding to Fig. 3g. **e**, eS24 blots corresponding to Fig. 4e. Blots represent at least two independent replicates (see "Statistics and Reproducibility" section). See Supplementary Fig. 10 for source data.

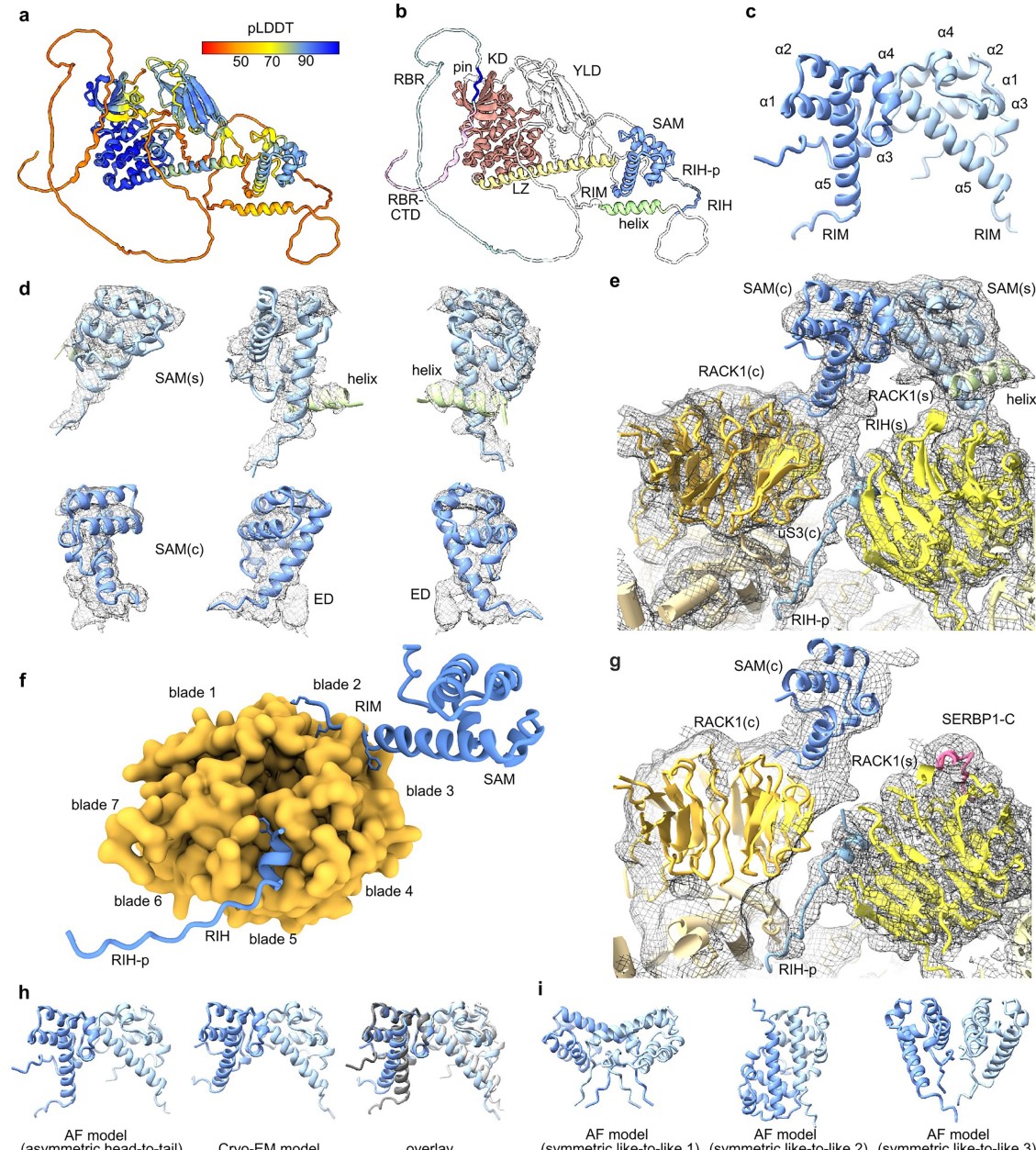

**Extended Data Fig. 4 | AlphaFold models and density fits for ZAKα.**
**a**, AlphaFold model of ZAKα, colored according to a per-model confidence
score (pLDDT; from 0 to 100). Blue regions display a very high confidence
(pLDDT > 90), red/orange region low confidence (pLDDT < 50). **b**, AlphaFold
model of ZAKα with individual domains color coded. Regions not visible in our
cryo-EM maps are shown in transparent grey except for KD (red), LZ (yellow),
RBR (pale blue) and CTD (plum). KD = kinase domain; LZ = leucine zipper; RBR =
ribosome binding region; CTD = C-terminal domain; RIM = RACK1-interacting
motif; RIH = RACK1-interacting helix; RIH-p = RIH-peptide; pin = eS27-pin;
YLD = YEATS-like domain; SAM = sterile alpha motif; helix = α-helix formed by
residues 568–583 of ZAK. **c**, AlphaFold model of a SAM dimer with asymmetric
head-to-tail interface. α-helices α1-5 are indicated. **d**, Three views on the
AlphaFold model for SAM(s) (top) and SAM(c) fitted into respective isolated
densities. The densities (mesh) were extracted from the local refined stalled
and collided 80S (see also Extended Data Fig. 2) and were Gaussian low-pass

filtered with a standard deviation of 0.75 in ChimeraX 1.9. ED indicates low-
resolution density visible on SAM(c) possibly representing the same helix or
a sequence preceding the RIH on the stalled ribosome. **e**, Cryo-EM map (grey
mesh) of stable disome the locally refined on the RACK1-SAM region (see also
Extended Data Fig. 2) with model for the ZAK-disome fitted. Extra density for
the ZAK RIH-p extends from RACK1(s) towards the disome interface. **f**, Model
of RACK1 shown as surface with bound ZAK RIH/RIH-p and with ZAK RIM/SAM.
**g**, Cryo-EM map of the stalled 80S derived from the native pull-down of kinase
inactive K394D ZAK ribosomal complexes, locally refined around the RACK1-
SAM region (see also Extended Data Fig. 9). Note that extra density was present
accounting for only one SAM domain emerging from the collided 80S. RACK1(s)
was bound to ZAK RIH with the RIH-p extending towards the collision interface
and to the C-terminus of SERBP1 (SERBP1-C). **h**, comparison of the SAM dimer
AlphaFold model (shown in grey in the overlay) with the cryo-EM derived model.
**i**, Three different symmetric like-to-like models of a ZAK dimer predicted by AF.

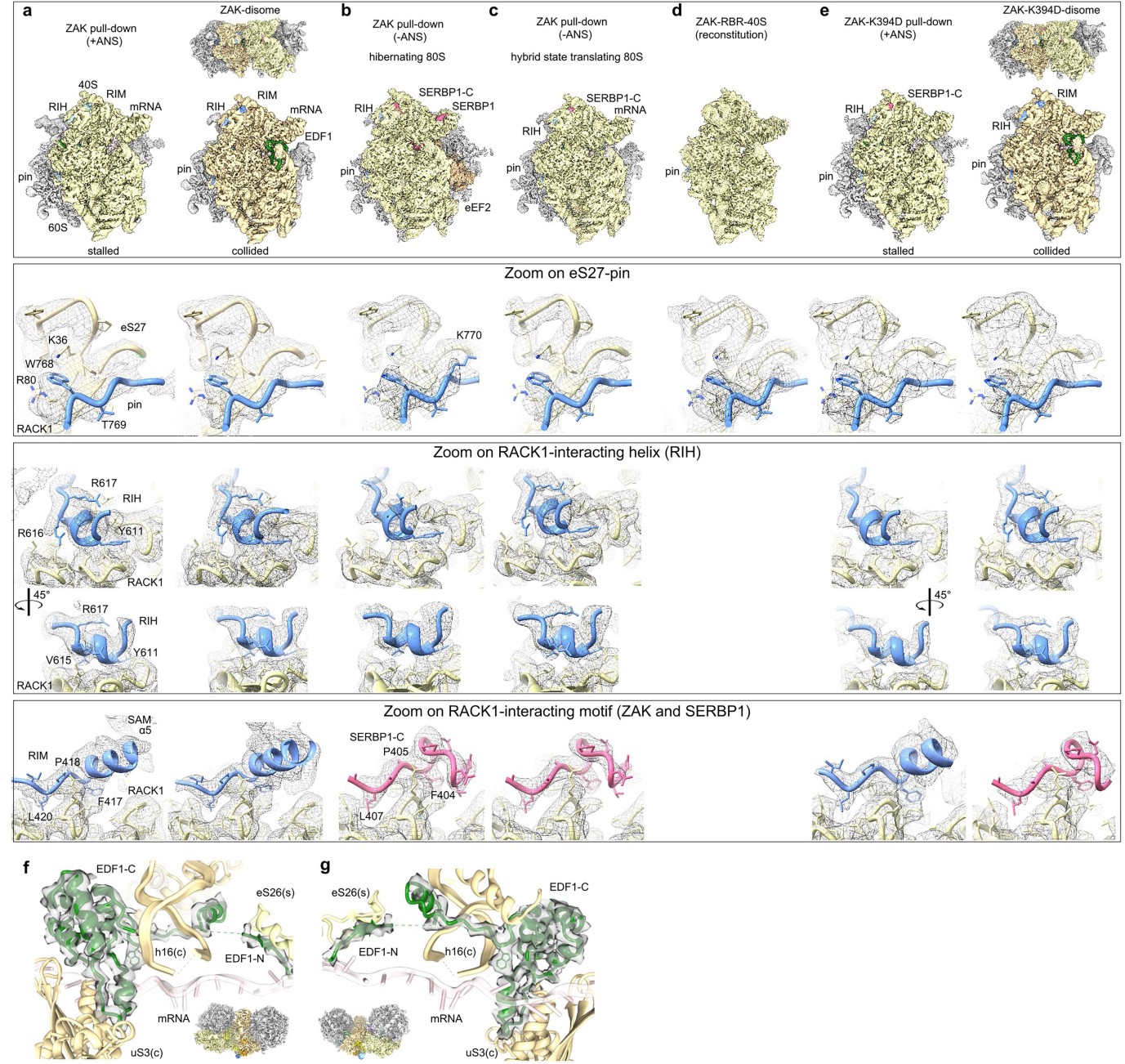

**Extended Data Fig. 5 | Fitting of ZAK eS27 pin, RIH, RIM and SERBP1-C and model of EDF1 bound to the colliding disome. a,** Top rows show cryo-EM maps of the individually refined stalled and collided 80S (Extended Data Fig. 2) from the dataset obtained from the kinase inactive ZAK pull-down after ANS treatment (+ANS). Below a view focusing on the eS27-pin, two views on RACK1-bound RIH and one view on the RACK1-bound RIM (with SAM α5) are shown. **b, c,** same as (**a**) for the untreated (-ANS) kinase inactive ZAK pull-down. Shown are cryo-EM maps for the refined hibernating (**b**) and hybrid state translating (**c**) 80S (see also Extended Data Fig. 3a). Views focusing on the ZAK eS27-pin, RIH and SERBP1-C (at the same RACK1 binding site as the ZAK RIM) and are shown as in (**a**). **d,** Cryo-EM map and view focusing on the eS27-pin for the reconstituted ZAK-RBR-40S complex. **e,** same as (**a**) with cryo-EM maps obtained from the

kinase inactive ZAK-K394D pull-down after ANS treatment dataset (Extended Data Fig. 9). For all density snapshots focusing on RACK1 the cryo-EM maps are derived from local refinements on the RACK1-SAM region (see also Extended Data Figs. 2, 3a and 9). For (**a**) and (**e**) composite disome maps are shown with hallmarks indicated (density for ZAK SAM domains omitted). All views show either the ZAK-disome model or the model containing RACK1, the ZAK RIH and SERBP1-C fitted into the maps (mesh); pin = eS27-pin; RIM = RACK1-interacting motif; RIH = RACK1-interacting helix; SAM = sterile alpha motif; RBR = ribosome binding region. **f, g,** Two views focusing on the EDF1 binding site. The molecular model for EDF1 is fitted into the isolated map, extracted from the composite map of the ZAK-disome complex. Thumbnails at the bottom indicate the view, boxes highlight the zoomed region.

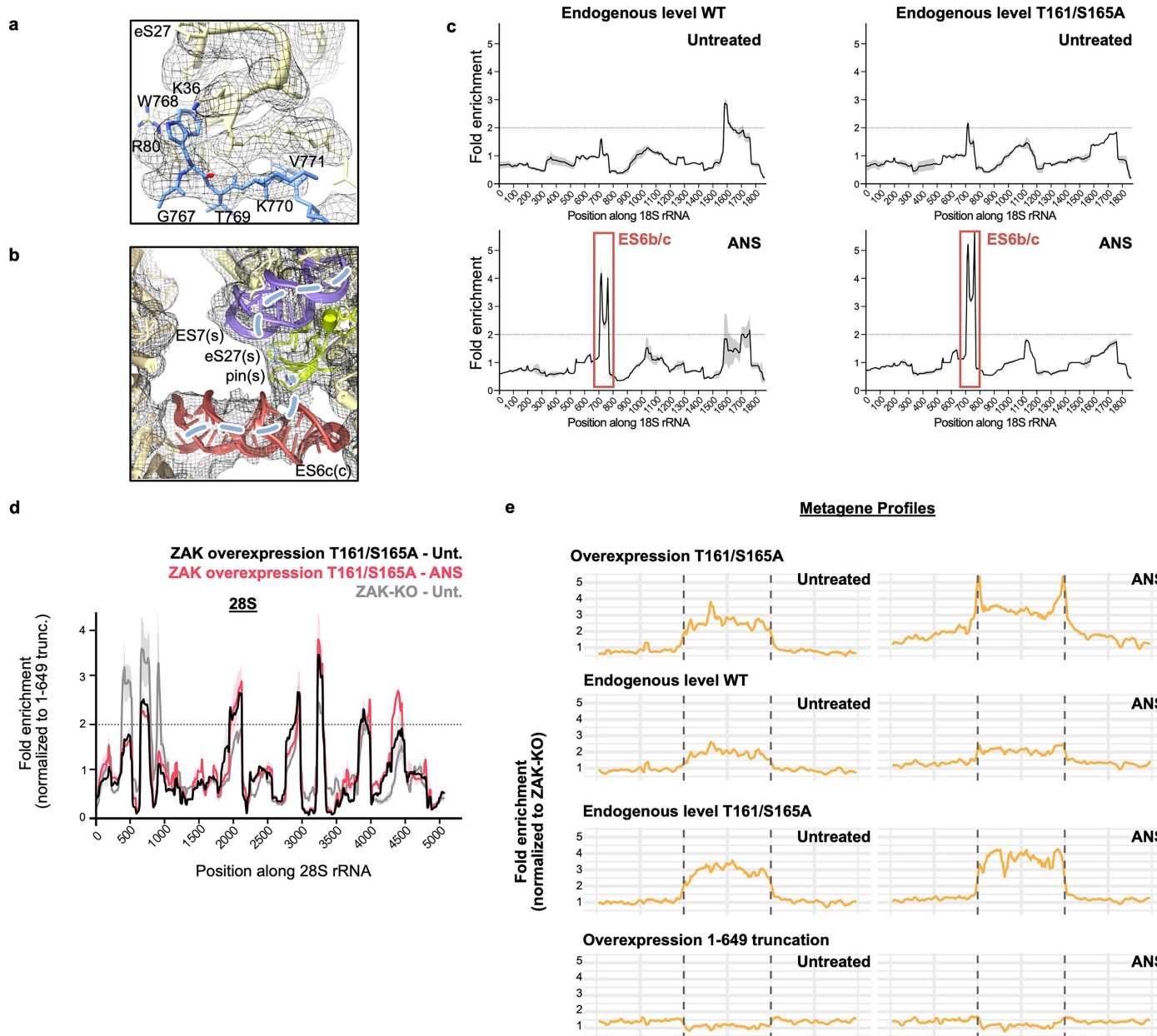

**Extended Data Fig. 6 | C-terminus structural validation and CLIP-seq analysis. a**, Model of the eS27-pin fitted into the 2.3 Å cryo-EM map (mesh) of reconstituted ZAK-RBR-40S complex (see also Extended Data Fig. 5d). The main interaction with eS27 is mediated by W768, which intercalates between R80 and K36 of eS27. The following residues (T769, K770 and V771) are packed on top of the 3-stranded β-sheet of eS27 with T769 and V771 forming additional interactions with eS27. **b**, View focusing on the collision interface of the ZAK-disome complex surrounding the eS27-pin. The model for the ZAK-disome was fitted into the cryo-EM map of the entire stable ZAK-disome (Extended Data Fig. 2, right side). Dashed lines indicate regions flanking the eS27-pin. **c**, CLIP-seq of Flag tagged WT and T161A/S165A ZAK expressed from partial CMV promoter in HEK293T ZAK-KOs and its associated ribosomal RNA (black lines) in untreated and ANS treated conditions. Sequence reads were mapped

to 18S rRNA and normalized to non-ribosome binding 1–649 aa ZAK truncation sample. The horizontal line represents 2-fold enrichment. Shading indicates standard error of the mean from two biological replicates. Reads from CLIP-seq were mapped to the 18S rRNA and matched to regions including ES6b/c (red). **d**, CLIP-seq of T161A/S165A ZAK expressed from full CMV promoter (Black = overexpression, untreated; red = overexpression, ANS treated) and the ZAK-KO with no transfection (Grey = ZAK-KO, untreated). Sequence reads were mapped to 28S rRNA and normalized to 1–649 aa C-terminal truncation control. The horizontal line represents 2-fold enrichment. Shading indicates standard error of the mean from two biological replicates. **e**, Metagene analysis of each CLIP-seq sample (normalized to ZAK-KO). CLIP-seq coverage data was aligned to 5'UTR, CDS, and 3'UTR for each mRNA. Only ENSEMBL canonical transcripts were used.

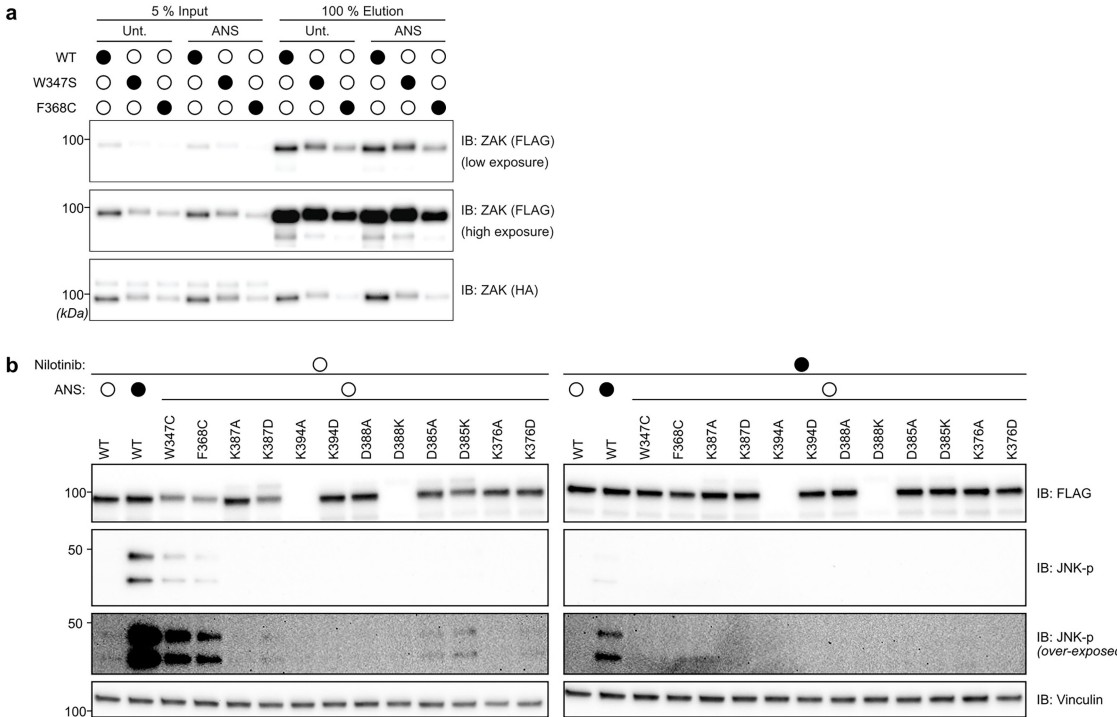

**Extended Data Fig. 7 | Characterization of ZAK interactions in vivo and in vitro. a**, Immunoblots of co-immunoprecipitation of FLAG tagged or HA tagged ZAK either in untreated or ANS treated conditions. **b**, Immunoblots of total lysate from HEK293T ZAK knockout cells transfected with partial CMV promoter plasmids expressing various SAM mutant N-terminal FLAG ZAK constructs at endogenous levels. Blots represent at least two independent replicates (see "Statistics and Reproducibility" section). See Supplementary Fig. 11 for source data.

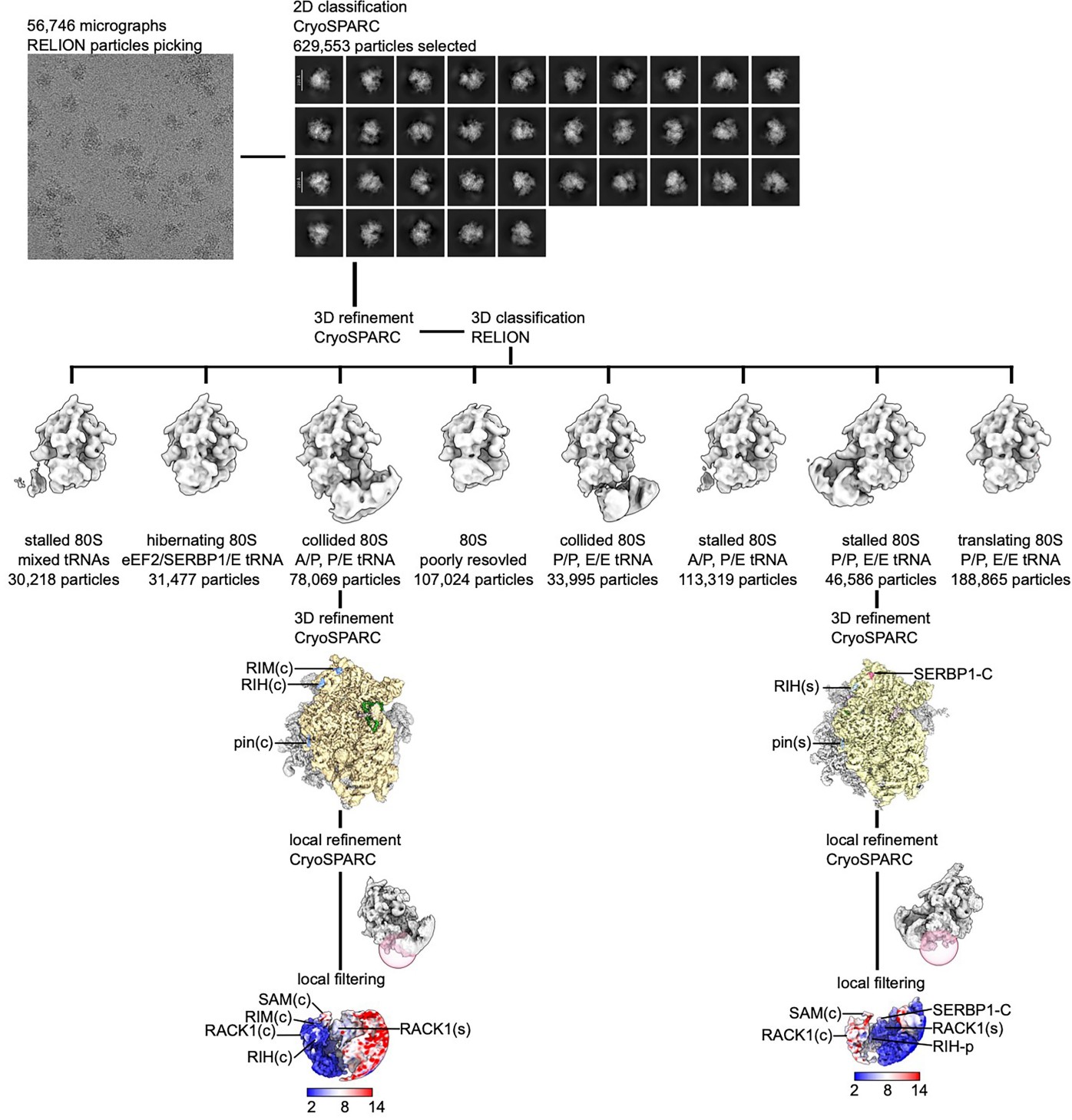

**Extended Data Fig. 8 | Cryo-EM data analysis, classification and resolution of the ZAK-K394D-disome complex.** The sample was obtained from a native pull-down using kinase inactive (K45M) ZAK-K394D mutant after ANS addition (see Extended Data Fig. 1b, right panel). 3D classification of 629,553 particles picked from a total of 56,746 micrographs showed 80S classes as described for the pull-downs using kinase inactive ZAK mutants described above. Amongst them were classes representing disomes as well as hibernating (with eEF2/SERBP1 and E tRNA) and translating (hybrid and POST state) 80S. The classes

representing stalled (with P/P and E/E tRNAs) and collided (with A/P and P/E tRNAs) 80S were refined followed by local refinement focusing on the RACK1-SAM region. Local resolution was determined for this region ranging from below 3 Å close to RACK1 to 5–15 Å for peripheral regions. We observed density for one SAM globular domain emerging from RACK1 of the collided ribosome, whereas the SERBP1 C-terminus (SERBP1-C) and the ZAK RIH occupy RACK1 of the stalled ribosome.

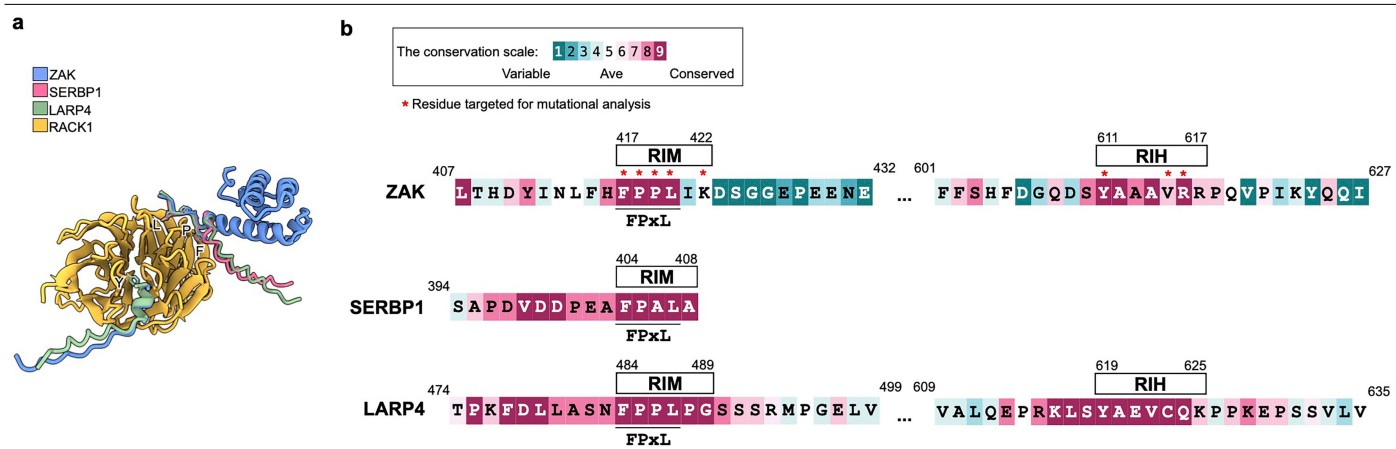

**Extended Data Fig. 9 | Comparison of RACK1-bound ZAK, SERBP1 and LARP4.**
**a**, Overlay of AlphaFold models for RACK1-bound ZAK (RIH, RIM and SAM domain), SERBP1-C and LARP4. Important conserved residues (F, P and L for the RIM and Y for the RIH) are indicated. **b**, Consurf evolutionary conservation profile RIM and RIH of ZAK with conserved SERBP1-C and LARP4 sequences aligned. Residues with red * were targeted for mutational analysis. ZAK RIM, SERBP1-C and LARP4 share the FPxL sequence for binding RACK1 and ZAK and LARP4 share a conserved Y residue in the RIH pinning it to RACK1.

# Extended Data Table 1 | Cryo-EM data collection, refinement and model validation

| | ZAK-bound human disome (EMD-54172) (PDB 9RPV) | SERBP1/RIH-bound RACK1 (EMD-54236) (PDB 9RSX) | Stalled 80S (kinase inactive ZAK pull-down with ANS treatment) (EMD-54140) | Collided 80S (kinase inactive ZAK pull-down with ANS treatment) (EMD-54141) | Stalled 80S focusing on ZAK-RACK1 (kinase inactive ZAK pull-down with ANS treatment) (EMD-54148) | Collided 80S focusing on ZAK-RACK1 (kinase inactive ZAK pull-down with ANS treatment) (EMD-54147) |
|---|---|---|---|---|---|---|
| **Data collection and processing** | | | | | | |
| Magnification | | 165000 | 165000 | 165000 | 165000 | 165000 |
| Voltage (kV) | | 300 | 300 | 300 | 300 | 300 |
| Electron exposure (e–/Å$^2$) | | 40 | 40 | 40 | 40 | 40 |
| Defocus range (μm) | | -0.5 to -3.5 | -0.5 to -3.5 | -0.5 to -3.5 | -0.5 to -3.5 | -0.5 to -3.5 |
| Pixel size (Å) | | 0.727 | 1.16 | 1.16 | 1.16 | 1.16 |
| Symmetry imposed | | C1 | C1 | C1 | C1 | C1 |
| Initial particle images (no.) | | 325,370 | 1,373,448 | 1,373,448 | 1,373,448 | 1,373,448 |
| Final particle images (no.) | | 108,673 | 123,142 | 139,996 | 123,142 | 139,996 |
| Map resolution (Å) | | 2.91 | 2.35 | 2.35 | 3 | 3 |
| FSC threshold | | 0.143 | 0.143 | 0.143 | 0.143 | 0.143 |
| | | | | | | |
| **Refinement** | Composite map | | | | | |
| Initial model used (PDB code) | 6Y57, 8GLP,7QVP, 6ZVH, AlphaFold | AlphaFold | | | | |
| Model resolution (Å) | 2.84 | 3.0 | | | | |
| FSC threshold | 0.5 | 0.5 | | | | |
| Model composition | | | | | | |
| Non-hydrogen atoms | 428,970 | 2734 | | | | |
| Protein residues | 23,238 | 353 | | | | |
| Ligands | 11,417 | --- | | | | |
| B factors (Å$^2$) | | | | | | |
| Protein | 0.00/8227.64/111.72 | 0.00/100.26/15.56 | | | | |
| Ligand | 25.22/238.69/67.49 | --- | | | | |
| R.m.s. deviations | | | | | | |
| Bond lengths (Å) | 0.006 (1) | 0.003 (0) | | | | |
| Bond angles (°) | 0.892 (642) | 0.510 (3) | | | | |
| Validation | | | | | | |
| MolProbity score | 1.61 | 1.59 | | | | |
| Clashscore | 3.71 | 4.81 | | | | |
| Poor rotamers (%) | 1.89 | 1.98 | | | | |
| Ramachandran plot | | | | | | |
| Favored (%) | 96.35 | 97.39 | | | | |
| Allowed (%) | 3.60 | 2.61 | | | | |
| Disallowed (%) | 0.04 | 0.00 | | | | |

| | Human disome with ZAK (kinase inactive ZAK pull-down with ANS treatment) (EMD-54191) | Hybrid state translating 80S (kinase inactive ZAK pull-down) (EMD-54149) | Hibernating 80S (kinase inactive ZAK pull-down) (EMD-54150) | ZAK-RBR-40S (in vitro reconstitution) (EMD-54165) | Stalled 80S (ZAK-K45M-K394D pull-down with ANS treatment) (EMD-54166) | Collided 80S (ZAK-K45M-K394D pull-down with ANS treatment) (EMD-54167) |
|---|---|---|---|---|---|---|
| **Data collection and processing** | | | | | | |
| Magnification | 165000 | 165000 | 165000 | 165000 | 165000 | 165000 |
| Voltage (kV) | 300 | 300 | 300 | 300 | 300 | 300 |
| Electron exposure (e–/Å$^2$) | 40 | 40 | 40 | 40 | 40 | 40 |
| Defocus range (μm) | -0.5 to -3.5 | -0.5 to -3.5 | -0.5 to -3.5 | -0.5 to -3.5 | -0.5 to -3.5 | -0.5 to -3.5 |
| Pixel size (Å) | 2.91 | 0.727 | 0.727 | 0.727 | 1.454 | 1.454 |
| Symmetry imposed | C1 | C1 | C1 | C1 | C1 | C1 |
| Initial particle images (no.) | 1,373,448 | 325,370 | 325,370 | 873,389 | 629,553 | 629,553 |
| Final particle images (no.) | 16,939 | 108,673 | 60,518 | 629,046 | 46,486 | 78,096 |
| Map resolution (Å) | 5.97 | 2.17 | 2.37 | 2.35 | 2.95 | 2.95 |
| FSC threshold | 0.143 | 0.143 | 0.143 | 0.143 | 0.143 | 0.143 |

| | Dr. Rachel Green |

# Reporting Summary

## Statistics

For all statistical analyses, confirm that the following items are present in the figure legend, table legend, main text, or Methods section.

| n/a | Confirmed | |
|---|---|---|
| ☐ | ☒ | The exact sample size (*n*) for each experimental group/condition, given as a discrete number and unit of measurement |
| ☐ | ☒ | A statement on whether measurements were taken from distinct samples or whether the same sample was measured repeatedly |
| ☐ | ☒ | The statistical test(s) used AND whether they are one- or two-sided *Only common tests should be described solely by name; describe more complex techniques in the Methods section.* |
| ☒ | ☐ | A description of all covariates tested |
| ☐ | ☒ | A description of any assumptions or corrections, such as tests of normality and adjustment for multiple comparisons |
| ☐ | ☒ | A full description of the statistical parameters including central tendency (e.g. means) or other basic estimates (e.g. regression coefficient) AND variation (e.g. standard deviation) or associated estimates of uncertainty (e.g. confidence intervals) |
| ☒ | ☐ | For null hypothesis testing, the test statistic (e.g. *F*, *t*, *r*) with confidence intervals, effect sizes, degrees of freedom and *P* value noted *Give P values as exact values whenever suitable.* |
| ☒ | ☐ | For Bayesian analysis, information on the choice of priors and Markov chain Monte Carlo settings |
| ☒ | ☐ | For hierarchical and complex designs, identification of the appropriate level for tests and full reporting of outcomes |
| ☒ | ☐ | Estimates of effect sizes (e.g. Cohen's *d*, Pearson's *r*), indicating how they were calculated |

*Our web collection on statistics for biologists contains articles on many of the points above.*

## Software and code

Policy information about availability of computer code

| Data collection | Cryo-EM data were collected with EPU v.3.7 software. |
|---|---|
| Data analysis | Western blots and coomassie gels were processed using: Image J 2.3. CLIP-seq data was processed using: mi_tools version 1.1.6; trim_galore version 0.6.10; STAR version 2.7.3a; samtools version 1.9; R version 4.4.1. Cryo-EM data was processed using: MotionCor2 (version 1.4.0), CTFFIND4 (version 4.1.13), RELION (version 5.0 beta), CryoSPARC (version 4.6.0). Molecular models were built and refined using WinCoot 0.9.8.93 and Phenix1.20.1-4487. AlphaFold database (https://alphafold.ebi.ac.uk/), AlphaFold3 and AlphaFold2 Multimer were used for initial model prediction and its multimer implementation to analyze protein-protein interactions. Structural figures were created using Chimera X v1.9. |

For manuscripts utilizing custom algorithms or software that are central to the research but not yet described in published literature, software must be made available to editors and reviewers. We strongly encourage code deposition in a community repository (e.g. GitHub). See the Nature Portfolio guidelines for submitting code & software for further information.

## Data

Policy information about availability of data

All manuscripts must include a data availability statement. This statement should provide the following information, where applicable:
- Accession codes, unique identifiers, or web links for publicly available datasets
- A description of any restrictions on data availability
- For clinical datasets or third party data, please ensure that the statement adheres to our policy

The CLIP-seq data generated in this study have been deposited in the Gene Expression Omnibus (GSE299329). The cryo-EM structural data generated in this study have been deposited in the Electron Microscopy Data Bank and the Protein Data Bank (PDB), respectively, under the following accession codes: EMD-54172 for the composite ZAK-disome (obtained from kinase inactive ZAK pull-down with ANS treatment, PDB accession 9RPV); EMD-54140 for the stalled 80S and EMD-54141 for the collided 80S (related to composite map); EMD-54148 for the locally refined ZAK-RACK1 region of the stalled 80S and EMD-54147 for the locally refined ZAK-RACK1 region of the collided 80S (both related to composite map); EMD-ZZZZZ for the EMD-54149 for the hybrid state translating 80S and EMD-54150 for the hibernating 80S (obtained from native FLAG-ZAKα-K45M pull-down); EMD-54236 for the locally refined hybrid state translating 80S (PDB accession 9RSX); EMD-54165 for the in vitro reconstituted ZAK-RBR-40S complex; EMD-54166 for the stalled 80S and EMD-54167 for the collided 80S (obtained from FLAG-ZAKα-K45M K394D pull-down with ANS treatment). The structures used for atomic model building of ZAK-bound disome complexes are available from Worldwide Protein Data Bank (wwPDB) with accession codes 6Y57, 7QVP, 8GLP. All other data are presented in the main text of the manuscript (Figures 1-5) as well as supplemental (Extended Figures 1-9 and Supplemental Table 1 and Supplementary Information Documents).

## Research involving human participants, their data, or biological material

Policy information about studies with human participants or human data. See also policy information about sex, gender (identity/presentation), and sexual orientation and race, ethnicity and racism.

| Reporting on sex and gender | n/a |
|---|---|
| Reporting on race, ethnicity, or other socially relevant groupings | n/a |
| Population characteristics | n/a |
| Recruitment | n/a |
| Ethics oversight | n/a |

Note that full information on the approval of the study protocol must also be provided in the manuscript.

# Field-specific reporting

Please select the one below that is the best fit for your research. If you are not sure, read the appropriate sections before making your selection.

☒ Life sciences ☐ Behavioural & social sciences ☐ Ecological, evolutionary & environmental sciences

For a reference copy of the document with all sections, see nature.com/documents/nr-reporting-summary-flat.pdf

# Life sciences study design

All studies must disclose on these points even when the disclosure is negative.

| Sample size | For biochemistry and cell-based assays, no statistical methods or calculations were utilyzed. All experimental results were successfully replicated at least twice. The number of biological replicates was a minimum of two per assay. See "Statistics and Reproducibility" section for complete information. For Cryo-EM datasets sufficient number of micrographs were collected to achieve the reported resolution of map. Complete Cryo-EM statistics are provided in Extended Data Table 1. |
|---|---|
| Data exclusions | During cryo-EM data processing, particles were excluded if 2D class averages represented noise or did not show clearly identifiable features for 80S particles. |
| Replication | See Methods and the section titled "Statistics and Reproducibility" for details on replications. As discussed, all experiments were replicated on different days (biological replicates) in the same or similar experiments. All cryo-EM structures were determined from independent half datasets, which were compared to assess the resolution of the reconstruction. |
| Randomization | Randomization does not apply for biochemistry and cell-based assays in this project. To determine the overall resolution for cryo-EM reconstructions the "Gold standard" Fourier shell correlation (FSC) was used with a 0.143 cutoff criterion. Here, particle sets are randomly divided to generate two independent 3D maps that were used to calculate the FSC. |
| Blinding | No blinding was required for the reported experiments and therefore not attempted. |

# Reporting for specific materials, systems and methods

We require information from authors about some types of materials, experimental systems and methods used in many studies. Here, indicate whether each material, system or method listed is relevant to your study. If you are not sure if a list item applies to your research, read the appropriate section before selecting a response.

## Materials & experimental systems

| n/a | Involved in the study |
|-----|----------------------|
| ☐ | ☒ Antibodies |
| ☐ | ☒ Eukaryotic cell lines |
| ☒ | ☐ Palaeontology and archaeology |
| ☒ | ☐ Animals and other organisms |
| ☒ | ☐ Clinical data |
| ☒ | ☐ Dual use research of concern |
| ☒ | ☐ Plants |

## Methods

| n/a | Involved in the study |
|-----|----------------------|
| ☒ | ☐ ChIP-seq |
| ☒ | ☐ Flow cytometry |
| ☒ | ☐ MRI-based neuroimaging |

## Antibodies

| | |
|---|---|
| Antibodies used | Primary antibodies used:<br>Rabbit anti-eS24 (Abcam, Cat# ab196652, 1:1000)<br>Mouse anti-FLAG (Sigma, Cat#A8592, 1:5000)<br>Rabbit anti-HA (Cell Signaling, Cat#3724, 1:1000)<br>Mouse anti-JNK1 (Cell Signaling, Cat #3708, 1:1000)<br>Rabbit anti-Phospho-SAPK/JNK (Cell Signaling, Cat# 4668S, 1:1000)<br>Rabbit anti-RACK1 (Cell Signaling, Cat# 5432S, 1:1000)<br>Rabbit anti-SERBP1 (Novus, Cat# NBP1-85660, 1:1000)<br>Mouse anti-STREP (Sigma, Cat# 71591-3, 1:5000)<br>Mouse anti-Vinculin (Santa Cruz, Cat# sc-73614, 1:2000)<br>Rabbit anti-ZAK (Fortis, Cat# A301-993A, 1:1000)<br><br>Secondary antibodies used:<br>anti-Mouse (Cell Signaling, Cat# 7076S, 1:5000)<br>anti-Rabbit (Cell Signaling, Cat# 7074S, 1:5000) |
| Validation | anti-eS24 - Validated for IB by manufacturer, by publications listed on manufacturer website<br>anti-FLAG - Validated for IB by manufacturer, by experiments in Green lab (unpublished)<br>anti-HA - Validated for IB by manufacturer using SimpleChIP® Enzymatic Chromatin IP Kits<br>anti-JNK1 - Validated for IB by manufacturer, by publications listed on manufacturer website<br>anti-Phospho-SAPK/JNK - Validated for IB by manufacturer, by publications listed on manufacturer website<br>anti-RACK1 - Validated for IB by manufacturer, and by knockout experiments in this manuscript<br>anti-SERBP1 - Validated for IB by manufacturer and by siRNA knockdown experiment in this manuscript<br>anti-STREP - Validated for IB by manufacturer, by publications listed on manufacturer website<br>anti-Vinculin - Validated for IB by manufacturer, by publications listed on manufacturer website<br>anti-ZAK - Validated for IB by manufacturer and by knockout experiments in this manuscript (and unpublished)<br>anti-Mouse - Validated for IB by manufacturer using CST primary antibodies<br>anti-Rabbit - Validated for IB by manufacturer using CST primary antibodies |

## Eukaryotic cell lines

Policy information about cell lines and Sex and Gender in Research

| | |
|---|---|
| Cell line source(s) | HEK293T cells (ATCC CRL-3216); EXPi293F cells (Thermo Fisher; A14527) |
| Authentication | ZAK knockout and RACK1 knockout were authenticated by sequencing and western blot. |
| Mycoplasma contamination | Cell lines were negative for mycoplasma |
| Commonly misidentified lines (See ICLAC register) | n/a |

## Plants

| | |
|---|---|
| Seed stocks | n/a |
| Novel plant genotypes | n/a |
| Authentication | n/a |

