## [Peer Review File · Nature]

ZAK Activation at the Collided Ribosome

Corresponding Author: Professor Roland Beckmann

Version 0:

Reviewer comments:

Referee #1

(Remarks to the Author)

ZAK is a central player in the ribotoxic stress response. Previous studies from the authors as well as others has suggested that ZAK monitors the translational state of the cell by interacting with translating ribosomes. Current models suggest that stalled ribosomes and/or ribosomal collisions are recognized by ZAK and promote ZAK activation via autophosphorylation. However, little is known about how ZAK interacts with the ribosome and how this interaction leads to ZAK activation. Here the authors use a combination of biochemistry and cryo-EM to provide the first structural insights into how ZAK interacts with monosomes and collided disomes, and how this interaction can lead to activation of ZAK.

This is an important study that reports the long-awaited structure of ZAK on the ribosome and beautifully synergizes the findings from the structure with technically well performed and controlled in vivo and in vitro biochemical assays. The findings provide insights into the regions of ZAK that are generally important for general binding and monitoring of ribosomes, such as the RIH and pin domains, as well as additional regions that specifically recognize stalled and collided ribosomes, such as the RIM domain, which in turn facilitate activation of ZAK via dimerization of the SAM domain. The authors also describe an interesting interplay between the hibernation factor SERBP1 and ZAK suggesting that the former is a negative regulator of the RSR. The manuscript also provides a mechanistic basis for disease-associated mutations in the SAM domain of ZAK. Overall, this is a beautifully presented manuscript that should be of extreme interest to the general scientific community, especially those working on translation regulation, stress responses and kinases.

While I have no major concerns, there are a few points that could be addressed that will hopefully improve the manuscript.

1. The local resolution of the cryo-EM map for ZAK appears to be relatively limited, presumably due to flexibility. However, the molecular models generated are guided nicely by alphafold derived models and predicted interactions, as well as biochemistry. Nevertheless, there are many places in the manuscript where extensive descriptions of interactions are provided as well as accompanying figures e.g. related to panels Fig. 2b, 3c, 4b,c and 5c,d. It is not clear whether the authors really have the resolution to report and described all these interactions? Or are these based on AlphaFold predictions? This should be made clear in the text and in the figures so that there are no misunderstandings. I note that it is written on p. 30, line 473 that “we do not have sufficiently high resolution to assign amino acids points of contact...”; maybe one should not be showing them then or at least acknowledging AF?

2. In Fig. 1b, the legend says that this is a composite cryo-EM map. The text states that the composite map was obtained by refined focussing on individual 80S and on the entire disome. Do the authors mean that individually refined stalled and collided 80S were then used to build a composite map by fitting into the full 80S disome map? Was there not flexibility between the stalled and collided ribosomes – if so, then how does one define the “correct” angle between the two 80S to be presented in Fig. 1b? Will the composite map be deposited or the individual maps in a way that they opened aligned as if they were a disome?

3. Fig. 2g is a little hard to interpret, especially given the rather vague text (lines 231-232) associated with it. Once one has digested the legend, one can understand that WT (black) was normalized to the C-terminal mutant (grey) but unfortunately the variability also appears to be in grey? And in the ANS graph, I cannot see the grey line for the C-terminal mutant, only grey for the variability? What is also confusing is that in Ext. Data Fig. 7c, there doesn't appear to be a peak for the ES7 with

the endogenous but only for the ES6ac upon ANS treatment?

4. There are a lot of abbreviations in this paper and on p. 13, line 190, the authors make GST-fusions with the RBR. Maybe it would be good to write again here ribosome binding region to remind the reader. Although it has been defined earlier, so have many other abbreviations, or at least write RBR of the ZAK α or (C-terminal region of ZAK) or something.

5. On p. 25, line 368, it is stated "we identified a network of salt bridges...". Did the authors identify them or did AlphaFold?

6. On p. 423 the authors make a cryo-EM reconstruction of the K45M ZAK with the K394D mutation. Maybe a good place to remind the reader what the K45M mutation is there for.

Some minor points:

1. Lines 126-127: ...sterile alpha motif is abbreviated SAM. YEATS-Like domain is YLD. Maybe either Sterile Alpha Motif or YEATS-like domain, or YEATS-Like Domain.

2. Line 141: The name for RACK1-interacting peptide (RIH-peptide) is somewhat confusing since it is not actually interacting with RACK1 but rather with uS3?

3. Line 151 states "the ZAK-bound disome in a typical arrangement as described" could be misinterpreted that there were already structures of ZAK-bound disomes. Perhaps best to qualify that the arrangement is similar to other disomes that were not bound by ZAK?

4. Lines 183 and 188: Maybe "independent of the translation state" sounds better?

5. Fig. 5f,g appear to be redundant. It is actually counterproductive to have them since it just confuses the reader as to what one should be looking for/at.

6. Ext. Data Fig. 5 has two views of the RIH whereas only one view of the Pin and RIM are shown. Perhaps add a little indicator that its another view/rotation etc otherwise the reader spends time trying to work out what the difference is.

7. Ext. Data Table 1 has missing EMD-XXXX for hybrid state.

Referee #2

(Remarks to the Author)

In this manuscript, Huso, Niu, and coworkers demonstrate how ZAK α binds to and is activated by colliding ribosomes during ribotoxic stress. The MAP3K kinase ZAK α plays a central role in coordinating cellular response to stress. ZAK α constitutively associates with ribosomes and is released upon activation following ribosome collisions. So far, its dynamic interactions with the ribosome make it challenging to dissect the molecular details that govern ZAK α function in the ribotoxic stress response (RSR).

Here, the authors used a combination of biochemical, structural, and cell biology approaches to address this problem and elucidate the mechanism of ZAK α activation on ribosomes. They used a kinase-inactive version of this enzyme to trap ZAK α dimers on ribosomes during collision and then determine the structure this complex using single-particle cryo-EM. The cryo-EM structures revealed ZAK α dimers bridging interactions between collided and non-collided ribosome states identifying key regions of ZAK α for ribosome binding and activation. Mutations in these regions led to reduced association with polysomes and diminished activation. Furthermore, a RACK1-interacting SAM domain was shown to mediate ZAK α dimerization on the ribosome and is essential for its activation and collision sensing. The authors propose an elegant model in which ZAK α functions as a "molecular tape measure" that senses inter-ribosomal distance during collisions, thereby modulating its activation.

Overall, the conclusions are well supported by rigorous and thoughtfully designed experiments, making the study suitable for publication in Nature. Several remaining comments are outlined below would further strengthen the model presented in the manuscript prior to publication:

1. Mutations in either the ES27 "pin" region or the RIH domain of ZAK α abolish ribosome binding and activation. It is somewhat surprising that no additive effect is observed, given that ZAK α contains multiple ribosome-binding regions. One might expect that mutating the RIH domain alone would reduce, but not abolish, ribosome binding and allow for some residual activation, since contacts via the ES27 pin region would still be intact. Unless an allosteric effect or perhaps a conformational change underlies ZAK α activation, an explanation in the Discussion would be helpful to clarify why single-point mutations in either region are sufficient to eliminate function.

2. The proposed competition between the ZAK RIM domain and SERBP1 for an overlapping binding site on RACK1 is an interesting observation. However, if knockdown of SERBP1 leads to increased JNK phosphorylation, it is thus possible that ZAK overexpression will allow for displacement of SERBP1 from polysomes to free ribosomal fractions. Therefore, the authors should test SERBP1 levels in polysomes gradients when ZAK is activated (in the presence of anisomycin) and ZAK binding mutants to strengthen their proposed model.

3. The structure clearly shows that the SAM domains of ZAK α form a dimer interface on the ribosome, which appears to

contribute to its activation. However, it remains unclear whether ZAK α exists as a dimer in solution or forms dimers only upon binding to individual 80S ribosomes. The authors briefly touched on this point in the Introduction and Discussion, but a more detailed clarification would strengthen the model. One possibility is that ZAK α exists as an inactive dimer in the cytosol and is recruited to collided ribosomes, where binding induces a conformational rearrangement of the SAM domains from tail-to-tail to head-to-head orientation that facilitates kinase activation. Was the oligomeric state of free ZAK tested? Expanding on this model and its supporting evidence would help contextualize the structural findings within the broader framework of ZAK α regulation.

4. There is additional density to the right of the ZAK α SAM domain depicted in Fig. 5f. Can the authors comment whether this density can be unambiguously assigned and if so, can they update their model?

5. The schematic in Fig. 6 is difficult to interpret in its current form. In particular, the pre-activation panel includes features that cannot be clearly viewed by readers. Additionally, it is unclear whether the figure is meant to depict collided ribosomes or a ZAK α dimer bound to two separate translating ribosomes prior to collision. Should it there two RACK1 one on each ribosome in the first panel? Clarifying these elements either through improved labeling, scaling, or separation of panels would greatly enhance the reader's understanding of the proposed activation models.

Minor Comments:

- In Fig. 1e, a low-pass filtered envelope of the ZAK α disome is shown. The authors note that this region is flexible and cannot be unambiguously built, and it is excluded from the final deposited atomic model. Given the low confidence in the fit and the limited contribution to the results in Fig. 1, this panel could be removed from the main figures and placed in the supplement, if necessary.
- On line 331, the study referenced a recent cryo-electron tomography study as having visualized SERBP1 density on native ribosomes. However, the correct technique used in this study is not cryo-ET but "in situ single-particle cryo-EM" done on the lamellae of cells. Please correct this.
- When analyzing ribosome distribution in cells overexpressing different ZAK α variants, was there a change in the overall distribution of monosomes and polysomes compared to wild-type? Several panels (e.g., Figs 2e, 3e,g, 4e) include es24 staining, but it's unclear what cell types these are from or how ribosome distribution compares across conditions. Including control blots, or showing a representative ribosome profile (e.g., from WT and a ZAK α mutant) would help contextualize these findings.
- The dimer interface between the two SAM domains contains several charged residues in close proximity that could form salt bridges. It is a bit surprising that mutations in two nearby lysines (K387 and K394), both located at the interface, appear to have opposite effects on kinase activity, can authors clarify/comment on this a bit more?

Referee #3

(Remarks to the Author)

I co-reviewed this manuscript with one of the reviewers who provided the listed reports.

Referee #4

(Remarks to the Author)

The manuscript "ZAK activation at the collided ribosome" presents new information on how the ZAK kinase, which is an important protein in the ribotoxic stress response (RSR), binds the colliding ribosome and is activated to regulate cell fate in response to RSR. Although previous reports have shown that ZAK kinase associates with the ribosome and is activated and released from the ribosome under stress-induced inhibition of protein synthesis, the mechanism of activation remains unknown. The authors here present compelling evidence that the ribosome is acting as a scaffold for recruitment and activation of the ZAK kinase. The studies are well done and supportive of this very exciting new mechanism. My comments below aim to improve the manuscript and strengthen their conclusion that ZAK kinase actions via the ribosome determine cell fate to RSR.

Comments

1. In Fig. 1a, the tracer of the gradient is from WT transfected cells. I suggest to also show in supplemental data the tracers of untransfected (WT), transfected active and transfected inactive ZAK. Are the tracers different? This is important because overexpression of ZAK activates it.
2. In Fig. 2e, it will help to evaluate the stress response by providing the distribution of eS24 for all conditions.
3. In Fig. 3e and other figures, ANS causes a partial release of ZAK from the ribosome. Because there is no quantification of the data (only trends of distribution), the authors should comment on this. Is it the relative % of colliding ribosomes during treatment? Can disomes identified by structural studies in Fig. 1 help explain the partial release? Also the smear of phosphor ZAK in the figure should be marked.
4. In all experiments that ZAK-KO cells are used to transfect ZAK and ZAK mutants to endogenous levels, it is not clear from

the Extended Data how the authors evaluated this. Did they compare levels of the ZAK proteins in WT, KO and transfected cells in parallel with an antibody against ZAK that recognizes endogenous and transfected proteins? It seems that the conclusion is drawn from relative levels between partial and complete expression ZAK constructs, which is not sufficient. Please explain.

5. In line 222, the authors state that loss of ribosome binding correlates with loss of ZAK activation and JNK phosphorylation. Is it because released ZAK cannot be autophosphorylated or this mutant cannot be autophosphorylated? I notice that in RACK1^{-/-} cells the endogenous ZAK protein is not phosphorylated when not bound to the ribosome. It requires a discussion.

6. The importance of the study also lies in the fact that ZAK kinase activation determines cell fate. The authors have previously shown that ribotoxic stress drives UV-mediated cell death. Can the authors use the reconstituted cells with the different mutant ZAK kinases (including the pathogenic ones) and determine tolerance (by washing-off ANS) or vulnerability by using UV or by other means that the authors wish to show the physiological significance of the structural/mechanistic findings?

Overall, it was a pleasure to read this manuscript. I believe the suggested revisions will broaden the significance of the findings.

Version 1:

Reviewer comments:

Referee #1

(Remarks to the Author)

The authors have addressed all my comments satisfactorily.

Referee #2

(Remarks to the Author)

The authors have addressed all of my concerns. I have no further suggestions and recommend the manuscript for publication.

Referee #3

(Remarks to the Author)

I co-reviewed this manuscript with one of the reviewers who provided the listed reports.

Referee #4

(Remarks to the Author)

I read the response to all reviewers and I found the revisions acceptable. The manuscript is suitable for publication in Nature.

Point-by-point response to the referees

Referee #1:

ZAK is a central player in the ribotoxic stress response. Previous studies from the authors as well as others has suggested that ZAK monitors the translational state of the cell by interacting with translating ribosomes. Current models suggest that stalled ribosomes and/or ribosomal collisions are recognized by ZAK and promote ZAK activation via autophosphorylation. However, little is known about how ZAK interacts with the ribosome and how this interaction leads to ZAK activation. Here the authors use a combination of biochemistry and cryo-EM to provide the first structural insights into how ZAK interacts with monosomes and collided disomes, and how this interaction can lead to activation of ZAK.

This is an important study that reports the long-awaited structure of ZAK on the ribosome and beautifully synergizes the findings from the structure with technically well performed and controlled in vivo and in vitro biochemical assays. The findings provide insights into the regions of ZAK that are generally important for general binding and monitoring of ribosomes, such as the RIH and pin domains, as well as additional regions that specifically recognize stalled and collided ribosomes, such as the RIM domain, which in turn facilitate activation of ZAK via dimerization of the SAM domain. The authors also describe an interesting interplay between the hibernation factor SERBP1 and ZAK suggesting that the former is a negative regulator of the RSR. The manuscript also provides a mechanistic basis for disease-associated mutations in the SAM domain of ZAK. Overall, this is a beautifully presented manuscript that should be of extreme interest to the general scientific community, especially those working on translation regulation, stress responses and kinases.

While I have no major concerns, there are a few points that could be addressed that will hopefully improve the manuscript.

1. The local resolution of the cryo-EM map for ZAK appears to be relatively limited, presumably due to flexibility. However, the molecular models generated are guided nicely by alphafold derived models and predicted interactions, as well as biochemistry. Nevertheless, there are many places in the manuscript where extensive descriptions of interactions are provided as well as accompanying figures e.g. related to panels Fig. 2b, 3c, 4b,c and 5c,d. It is not clear whether the authors really have the resolution to report and describe all these interactions? Or are these based on AlphaFold predictions? This should be made clear in the text and in the figures so that there are no misunderstandings. I note that it is written on p. 30, line 473 that “we do not have

sufficiently high resolution to assign amino acids points of contact...”; maybe one should not be showing them then or at least acknowledging AF?

Response:

We apologize for being somewhat unclear about the (local) resolution in the respective regions described in the above mentioned figure panels. For most ZAK-ribosome contacts (i.e., eS27-pin, RIH and RIM) we have sufficient local resolution to assign the amino acids shown in the panels of figures 2-4. To help assess the resolution of the various ZAK-ribosome contacts we provided Extended Data Figure 5 showing our molecular model fit into our densities. We revised this figure to make it now easier for the reader to relate the models shown in main figures to the actual densities they are based on. We now clearly state in the main text, which parts of the model are supported by side-chain level density and which parts are solely AF-based. Details relating to individual figure panels are below:

Fig. 2b shows the eS27-pin interaction with eS27. We see clear side chain density for W768 of the pin in all cryo-EM maps and K36/R80 of eS27 is best visible in the reconstituted GST-RBR-40S complex map, for which we reached a resolution of about 2.3 Å in the eS27 region. We demonstrate this in the revised Extended Data Fig. 5 (second row).

Fig. 3c shows the interaction between the ZAK RIH with RACK1. Clear density is visible for Y611 in all maps and for R616 and R617 in the individual reconstructions of the stalled and collided ribosomes of the ZAK-disome. This is shown in revised Extended Data Fig. 5, third row (two different views).

Fig. 4b and 4c show the interaction between the ZAK RIM with RACK1. Here, the FPxL motifs of both ZAK-RIM and SERBP1-C could be fitted based on resolved side chain density (Extended Data Fig. 5, fourth row).

Fig. 5c shows the AF-based model for the SAM-SAM dimer. Here, no side chain resolution is available. A fit into low-pass filtered density is shown in Extended Data Fig. 4d and 4e. We added a statement in the legend that the model shown in Fig. 5c is based on AlphaFold.

Finally, the RIH-peptide, extending from the RIH towards the ISS was not resolved at side chain level, as explained in on p. 30, line 473 of the original manuscript. Here, as also described in the methods section, we fitted a AlphaFold prediction of a ZAK residues 600-631 (containing the RIH and RIH-peptide) bound to RACK1 into still clearly visible continuous density (shown in Extended Data Fig. 4e)

2. In Fig. 1b, the legend says that this is a composite cryo-EM map. The text states that the composite map was obtained by refined focussing on individual 80S and on the

entire disome. Do the authors mean that individually refined stalled and collided 80S were then used to build a composite map by fitting into the full 80S disome map? Was there not flexibility between the stalled and collided ribosomes – if so, then how does one define the “correct” angle between the two 80S to be presented in Fig. 1b? Will the composite map be deposited or the individual maps in a way that they opened aligned as if they were a disome?

Response:

The reviewer is right, for generating the composite map stalled and collided ribosomes were individually refined to best-possible resolution and then fit into a map obtained from refining a class representing a complete stable disome, thereby providing the angle between the two individual ribosomes (see also Extended Data Figure 2)

When analyzing the disome classes, we indeed observed conformational variability between the two ribosome entities (see Extended Data Figure 2; many 80S classes show density for neighbors varying in appearance and angle relative to the main 80S). Here, to obtain molecular resolution of the ZAK-ribosome contact points, we decided to first individually refine 80S classes for stalled and collided ribosomes. To display them in context of a “true” stable disome, we followed the most common way to join maps “correctly”: this is to obtain a consensus map of the entire particle (here the ZAK-bound disome) and then fit the focused/locally refined maps (the 80S ribosomes and the locally refined ZAK densities) and combine them using the ChimeraX “vop max” function of similar tools in Phenix.

We chose a relatively well-resolved stable disome class (6 Angström) obtained after 3D classification as the best possible option for a consensus reconstruction of a ZAK-bound stable disome. This map matches all features of the two individually refined 80S even at the collision interface, but at lower resolution (see Extended Data Fig. 4e and Extended Data Fig. 7b).

We revised Extended Data Fig. 2 that now shows more clearly which disome map was used as consensus map and which maps are part of the composite map. Moreover, we explain in more detail in the Methods, how the composite map was obtained.

(Of note, the question how to represent combined cryo-EM maps after local/focused or Multi-Body refinements is controversially discussed in the cryo-EM field and there is no easy way to take such heterogeneity into account in one simple image)

We have deposited the composite map as well as the corresponding 80S maps with individual EMD entries. Unfortunately, the deposition system doesn't favor to upload the

aligned 80S maps, because it requires corresponding original “half maps” for an independent assessment of the maps’ resolution. All information of the high-resolution 80S maps, however, is preserved in the composite map and it is not absolutely necessary for the common user to align the individual maps.

3. Fig. 2g is a little hard to interpret, especially given the rather vague text (lines 231-232) associated with it. Once one has digested the legend, one can understand that WT (black) was normalized to the C-terminal mutant (grey) but unfortunately the variability also appears to be in grey? And in the ANS graph, I cannot see the grey line for the C-terminal mutant, only grey for the variability? What is also confusing is that in Ext. Data Fig. 7c, there doesn’t appear to be a peak for the ES7 with the endogenous but only for the ES6ac upon ANS treatment?

Response:

The grey line in the top panel of Fig. 2g was mis-labeled in the original figure legend. This grey line corresponds to the ZAK KO sample and is included to provide a reference for noise in the data (and therefore to help the reader appreciate that the ES7 and ES6b/c peaks correspond to a large signal above background). We can remove this ZAK KO line if needed, but we feel it adds to the interpretation of the data. The main text and legend have been updated to reflect these clarifications. We did not generate a ZAK KO ANS-treated sample and that is why there is no grey line in the bottom plot.

In regards to Ext. Data Fig 7c, the Reviewer is correct that we do not observe the ES7 peak with ZAK constructs expressed at endogenous levels. This is likely due to inherent noise with mapping sequencing data to rRNA; because rRNAs are highly abundant, there are reads which map across the entire 18S locus even in a ZAK KO or beads-only sample (and this why sequencing protocols usually incorporate an rRNA depletion). In the case of ES7, it seems to be a region of the 18S rRNA where reads resulting from noise outweigh reads resulting from biology (endogenous ZAK CLIPing to this site). Only with the overexpression of ZAK do we gain enough reads (above noise) to appreciate the peak.

4. There are a lot of abbreviations in this paper and on p. 13, line 190, the authors make GST-fusions with the RBR. Maybe it would be good to write again here ribosome binding region to remind the reader. Although it has been defined earlier, so have many other abbreviations, or at least write RBR of the ZAKa or (C-terminal region of ZAK) or something.

Response:

The text has been updated to include the following:

“GST-tagged RBR (ribosome binding region, the last 100 amino acids of ZAK including the eS27-pin)”

5. On p. 25, line 368, it is stated “we identified a network of salt bridges...”. Did the authors identify them or did AlphaFold?

Response:

We thank the reviewer for pointing this out. For the structural details of the SAM dimer interface and the identification of the salt bridges, we indeed used the AlphaFold model. We corrected this in the text and the figure legend.

6. On p. 423 the authors make a cryo-EM reconstruction of the K45M ZAK with the K394D mutation. Maybe a good place to remind the reader what the K45M mutation is there for.

Response:

The text has been updated to include the following:

“K45M ZAK (kinase inactive mutant used for previous structural pull-downs)”

Some minor points:

1. Lines 126-127: ...sterile alpha motif is abbreviated SAM. YEATS-Like domain is YLD. Maybe either Sterile Alpha Motif or YEATS-like domain, or YEATS-Like Domain.

Response:

Text has been updated. We kept sterile alpha motif lowercase but changed the YEATS-Like Domain to YEATS-like domain.

2. Line 141: The name for RACK1-interacting peptide (RIH-peptide) is somewhat confusing since it is not actually interacting with RACK1 but rather with uS3?

Response:

We agree that this may be confusing. However, to be clear, “RIH-peptide” would stand for “RACK1-interacting helix peptide”. We chose the name because it is a (C-terminal) extension from RIH (which interacts with RACK1).

3. Line 151 states “the ZAK-bound disome in a typical arrangement as described” could be misinterpreted that there were already structures of ZAK-bound disomes. Perhaps best to qualify that the arrangement is similar to other disomes that were not bound by ZAK?

Response:

We agree and clarified this point in the text, which now reads:

“The ZAK-bound disome was in a typical arrangement as described before for human disomes”

4. Lines 183 and 188: Maybe “independent of the translation state” sounds better?

Response:

We agree and updated the text in the revised version.

5. Fig. 5f,g appear to be redundant. It is actually counterproductive to have them since it just confuses the reader as to what one should be looking for/at.

Response:

The two panels intended to provide complementary information.

The original idea was to show the different densities for the SAM dimer and monomer, respectively, in Fig. 5f and the molecular model of the monomeric SAM in Fig. 5g. However, since the models are nicely recognizable in 5f already, we agree with the Reviewer and removed Fig. 5g from the revised manuscript.

6. Ext. Data Fig. 5 has two views of the RIH whereas only one view of the Pin and RIM are shown. Perhaps add a little indicator that its another view/rotation etc otherwise the reader spends time trying to work out what the difference is.

Response:

We agree that Ext. Data Figure 5 was somewhat hard to read. As already outlined in the answer to point 1, we revised this figure and indicate now that there are two views for the RIH.

7. Ext. Data Table 1 has missing EMD-XXXX for hybrid state.

Response:

All EMD entries are now present in revised Ext. Data Table 1.

Referee #2:

In this manuscript, Huso, Niu, and coworkers demonstrate how ZAK α binds to and is activated by colliding ribosomes during ribotoxic stress. The MAP3K kinase ZAK α plays a central role in coordinating cellular response to stress. ZAK α constitutively associates with ribosomes and is released upon activation following ribosome collisions. So far, its dynamic interactions with the ribosome make it challenging to dissect the molecular details that govern ZAK α function in the ribotoxic stress response (RSR).

Here, the authors used a combination of biochemical, structural, and cell biology approaches to address this problem and elucidate the mechanism of ZAK α activation on ribosomes. They used a kinase-inactive version of this enzyme to trap ZAK α dimers on ribosomes during collision and then determine the structure this complex using single-particle cryo-EM. The cryo-EM structures revealed ZAK α dimers bridging interactions between collided and non-collided ribosome states identifying key regions of ZAK α for ribosome binding and activation. Mutations in these regions led to reduced association with polysomes and diminished activation. Furthermore, a RACK1-interacting SAM

domain was shown to mediate ZAK α dimerization on the ribosome and is essential for its activation and collision sensing. The authors propose an elegant model in which ZAK α functions as a "molecular tape measure" that senses inter-ribosomal distance during collisions, thereby modulating its activation.

Overall, the conclusions are well supported by rigorous and thoughtfully designed experiments, making the study suitable for publication in Nature. Several remaining comments are outlined below would further strengthen the model presented in the manuscript prior to publication:

1. Mutations in either the ES27 "pin" region or the RIH domain of ZAK α abolish ribosome binding and activation. It is somewhat surprising that no additive effect is observed, given that ZAK α contains multiple ribosome-binding regions. One might expect that mutating the RIH domain alone would reduce, but not abolish, ribosome binding and allow for some residual activation, since contacts via the ES27 pin region would still be intact. Unless an allosteric effect or perhaps a conformational change underlies ZAK α activation, an explanation in the Discussion would be helpful to clarify why single-point mutations in either region are sufficient to eliminate function.

Response:

While we do observe a subtle additive effect on binding and activation when combining the eS27-pin and ES7-patch mutations (Fig. 2e and Fig. 2f), we agree that the strong loss of binding/activation phenotype observed with either the eS27-pin or the RIH mutant alone is interesting. In our native pull-downs, we observe that both the eS27-pin and RIH are present in all ZAK bound ribosomes (including monosomes and disomes). Indeed, our *in vitro* pull-down with purified GST-RBR (which contains only the eS27-pin) (Fig. 2) demonstrates that the eS27-pin is sufficient to bind ribosomes (both 80S and 40S), at least under our *in vitro* conditions. The discussion text has been updated regarding this point.

2. The proposed competition between the ZAK RIM domain and SERBP1 for an overlapping binding site on RACK1 is an interesting observation. However, if knockdown of SERBP1 leads to increased JNK phosphorylation, it is thus possible that ZAK overexpression will allow for displacement of SERBP1 from polysomes to free ribosomal fractions. Therefore, the authors should test SERBP1 levels in polysomes gradients when ZAK is activated (in the presence of anisomycin) and ZAK binding mutants to strengthen their proposed model.

Response:

While this is an interesting suggestion, after consideration, we think that while overexpression of ZAK may outcompete SERBP1 binding at the RIM (FPxL motif) site, this still may not cause loss of SERBP1 binding to polysomes since SERBP1 has multiple other points of contact with the ribosome that would not be disrupted. A recent publication characterizes SERBP1 binding to the 60S subunit via its N-terminus as well as extensive binding to the 40S subunit via its C-terminus (Zheng et al., 2024). We predicted that these multiple SERBP1:ribosome interactions (separate from its FPxL motif interaction) would likely not be disrupted by ZAK overexpression. As such, we chose to focus on the increases in ZAK activity that we report rather than on SERBP1 binding.

Zheng, W. *et al.* Visualizing the translation landscape in human cells at high resolution. *bioRxiv* 2024.07.02.601723 (2024) doi:10.1101/2024.07.02.601723.

3. The structure clearly shows that the SAM domains of ZAK α form a dimer interface on the ribosome, which appears to contribute to its activation. However, it remains unclear whether ZAK α exists as a dimer in solution or forms dimers only upon binding to individual 80S ribosomes. The authors briefly touched on this point in the Introduction and Discussion, but a more detailed clarification would strengthen the model. One possibility is that ZAK α exists as an inactive dimer in the cytosol and is recruited to collided ribosomes, where binding induces a conformational rearrangement of the SAM domains from tail-to-tail to head-to-head orientation that facilitates kinase activation. Was the oligomeric state of free ZAK tested? Expanding on this model and its supporting evidence would help contextualize the structural findings within the broader framework of ZAK α regulation.

Response:

We agree that the status of ZAK pre- and post-activation is an interesting question. Our co-immunoprecipitation assay in Extended Data Fig. 8a suggests that ZAK dimerizes to some extent pre-activation and that ANS treatment further increases the extent of dimerization. These data are consistent with the idea that both the colliding ribosome and 14-3-3 proteins (which bind activated ZAK in the cytosol post activation) act as scaffolds that promote/stabilize dimer formation. Also, as mentioned in the main text, the only published crystal structure of ZAK is the kinase domain dimer, again consistent with dimerization being critical to kinase activity (Mathea et al., 2016). The model that we favor is that ZAK is generally dimerized pre-activation, but in some auto-inhibited form, and that its binding to the colliding ribosome (at the RACK1 interface) relieves this inhibition. However, we are conservative in our discussion of this model as there is work

to be done to establish this conclusively. The text and Figure 6 have been updated to expand and clarify the model.

Mathea, S. *et al.* Structure of the human protein kinase ZAK in complex with vemurafenib. *ACS Chem. Biol.* **11**, 1595–1602 (2016).

4. There is additional density to the right of the ZAK α SAM domain depicted in Fig. 5f. Can the authors comment whether this density can be unambiguously assigned and if so, can they update their model?

Response:

We thank the reviewer for addressing this point. We are very intrigued by finding this extra density at the interface of RACK1 and the SAM domain of the collided ribosome. We also show this density in Extended Data Figure 4d, marked as “ED”. Unfortunately, all attempts to increase the local resolution at this site, including 3D variability analysis/3D flexible refinement/DynaMight failed to yield a map good enough to assign this density unambiguously. This unassigned extra density may be attributed to the “helix” formed by residues 569-583 of ZAK that packs against the SAM domain of the stalled ribosome; but it may also be the region N-terminal of the RIH (prior to residue 611) (see also Extended Data Fig. 4e for this).

5. The schematic in Fig. 6 is difficult to interpret in its current form. In particular, the pre-activation panel includes features that cannot be clearly viewed by readers. Additionally, it is unclear whether the figure is meant to depict collided ribosomes or a ZAK α dimer bound to two separate translating ribosomes prior to collision. Should it there two RACK1 one on each ribosome in the first panel? Clarifying these elements either through improved labeling, scaling, or separation of panels would greatly enhance the reader’s understanding of the proposed activation models.

Response:

We appreciate the feedback on this figure and have substantially improved the Figure 6 model (and the figure legend) to clarify these points.

Minor Comments:

- In Fig. 1e, a low-pass filtered envelope of the ZAK α disome is shown. The authors note that this region is flexible and cannot be unambiguously built, and it is excluded from the final deposited atomic model. Given the low confidence in the fit and the limited contribution to the results in Fig. 1, this panel could be removed from the main figures and placed in the supplement, if necessary.

Response:

We agree with the Reviewer and removed Fig. 1e from the main figures and placed it in Extended Data Fig 2.

- On line 331, the study referenced a recent cryo-electron tomography study as having visualized SERBP1 density on native ribosomes. However, the correct technique used in this study is not cryo-ET but “in situ single-particle cryo-EM” done on the lamellae of cells. Please correct this.

Response:

We apologize for being imprecise here. We corrected the text accordingly.

- When analyzing ribosome distribution in cells overexpressing different ZAK α variants, was there a change in the overall distribution of monosomes and polysomes compared to wild-type? Several panels (e.g., Figs 2e, 3e,g, 4e) include es24 staining, but it's unclear what cell types these are from or how ribosome distribution compares across conditions. Including control blots, or showing a representative ribosome profile (e.g., from WT and a ZAK α mutant) would help contextualize these findings.

Response:

The polysome profile from overexpression of WT ZAK in HEK293T cells is shown in Figure 1 (Fig. 1a) as a helpful visual/representative trace corresponding to the gradient fractions. The polysome profiles from untransfected and kinase inactive overexpression (T161/S165A) are included above. We observe an increase in the monosome peak for both WT and kinase inactive overexpression of ZAK compared to the untransfected control. We believe a large contributor to the increased monosome peak is transfection-related toxicity and is not therefore of biological interest here. It is also important to note that this overexpression system was only used for immunoprecipitation for cryo-EM or CLIP-seq; all other experiments that evaluated ZAK function were done with ZAK expressed at endogenous levels as outlined below.

Experiments mentioned in Figs. 2-5 were conducted in ZAK knockout HEK293T cells where ZAK is expressed from a transfected plasmid driven by a partial CMV promoter which allowed us to titrate ZAK expression down to endogenous ZAK levels (conditions established in Extended Data Fig. 1a). The eS24 blots for all gradients shown in Figs. 2-5 have been added to the extended data though we chose not to include them in the actual figures to simplify the presentation (Extended Data Fig. 3). All cell types used are noted in the Methods section. Importantly, we did not observe any appreciable changes in the polysome profiles when cells were transfected with different ZAK constructs expressed at endogenous levels using this partial promoter system; as such, we did not include traces in the manuscript. An example of this is included below:

- The dimer interface between the two SAM domains contains several charged residues in close proximity that could form salt bridges. It is a bit surprising that mutations in two nearby lysines (K387 and K394), both located at the interface, appear to have opposite effects on kinase activity, can authors clarify/comment on this a bit more?

Response:

We agree. The previously identified hyperactivating mutations (Fig. 5c) are located in the core of the SAM domain and AlphaFold modeling suggests that these hyperactive mutations disrupt its native fold and therefore lead to activation. Thus, this mode of activation is likely independent of SAM dimer formation on the ribosome and works by suppression of an auto-inhibitory function of the SAM domains which has been suggested earlier (Johansen et al., 2023). We presume that the different mutations at the interface (K387D and K394D) which cause diametrically opposite outcomes (hyper- and hypo-activation, respectively) must result in different conformational changes within the SAM domain, affecting either the proposed autoinhibiting activity (as for F368C) or the dimer formation capacity (as for K394D). We speculate that K387D may disrupt the SAM domain itself resulting in activation similar to hyperactive F368C by abolishing

autoinhibition. On the other hand, K394D, apparently abolishes the interface interaction for SAM dimer formation, and thus prevents (canonical) activation on the collided ribosome. As mentioned in the discussion, we are interested in further characterizing modes of ZAK kinase activation in future studies and we think our panel of identified SAM interface mutants offers a promising tool for these studies.

Johansen, V. B. I., Snieckute, G., Vind, A. C., Blasius, M. & Bekker-Jensen, S. Computational and functional analysis of structural features in the ZAK α kinase. *Cells* 12, 969 (2023).

Referee #3:

I co-reviewed this manuscript with one of the reviewers who provided the listed reports.

Referee #4:

The manuscript “ZAK activation at the collided ribosome” presents new information on how the ZAK kinase, which is an important protein in the ribotoxic stress response (RSR), binds the colliding ribosome and is activated to regulate cell fate in response to RSR. Although previous reports have shown that ZAK kinase associates with the ribosome and is activated and released from the ribosome under stress-induced inhibition of protein synthesis, the mechanism of activation remains unknown. The authors here present compelling evidence that the ribosome is acting as a scaffold for recruitment and activation of the ZAK kinase. The studies are well done and supportive of this very exciting new mechanism. My comments below aim to improve the manuscript and strengthen their conclusion that ZAK kinase actions via the ribosome determine cell fate to RSR.

Comments

1. In Fig. 1a, the tracer of the gradient is from WT transfected cells. I suggest to also show in supplemental data the tracers of untransfected (WT), transfected active and transfected inactive ZAK. Are the tracers different? This is important because overexpression of ZAK activates it.

Response:

As addressed above (Reviewer 2), the polysome profile traces from untransfected and kinase inactive ZAK are included below where we see a higher monosome peak for overexpression of both WT and kinase inactive ZAK compared to the untransfected control (all three traces shown below). In general, we think the major impact on the polysome profiles is from transfection toxicity and is not related to ZAK function or expression. Importantly, these overexpression conditions were only utilized for preparing complexes for cryo-EM or CLIP-seq; all analysis of ZAK function happened under conditions where ZAK expression was set at normal cellular levels (see Figures 2- 5).

2. In Fig. 2e, it will help to evaluate the stress response by providing the distribution of eS24 for all conditions.

Response:

The blots for eS24 are for each condition have been added to Extended Data Figure 3. As mentioned above (Reviewer 2, minor comment 3), we chose not to include them in the main figures since they did not vary for the different conditions and we felt made the figures more difficult to digest.

3. In Fig. 3e and other figures, ANS causes a partial release of ZAK from the ribosome. Because there is no quantification of the data (only trends of distribution), the authors should comment on this. Is it the relative % of colliding ribosomes during treatment? Can disomes identified by structural studies in Fig. 1 help explain the partial release? Also the smear of phosphor ZAK in the figure should be marked.

Response:

We have found that the amount of ZAK released from the ribosome is impacted by a number of factors. A primary variable is the timepoint post-activation (ANS treatment). For experiments conducted in this paper, we chose 15 minutes of anisomycin treatment as our collision-inducing condition based on previously published time courses optimizing ZAK activation and disome formation following ribotoxic stress (Sinha et al., 2025 as well as unpublished data from our lab). We note that our phos-tags of ZAK show that not all of ZAK is phosphorylated at 15 minutes (see Fig. 2f, 3h, 4f) suggesting that not all of ZAK is activated; this likely explains in part the remaining ZAK bound to polysomes. Further, we did not optimize these assays to look for ZAK release

downstream of activation but rather chose a convenient time point with relatively potent activation. Fig. 1a has been updated with phosphorylated ZAK highlighted.

Sinha, N. K. *et al.* The ribotoxic stress response drives UV-mediated cell death. *Cell* **187**, 3652–3670.e40 (2024).

4. In all experiments that ZAK-KO cells are used to transfect ZAK and ZAK mutants to endogenous levels, it is not clear from the Extended Data how the authors evaluated this. Did they compare levels of the ZAK proteins in WT, KO and transfected cells in parallel with an antibody against ZAK that recognizes endogenous and transfected proteins? It seems that the conclusion is drawn from relative levels between partial and complete expression ZAK constructs, which is not sufficient. Please explain.

Response:

We are happy to explain since this is an important point and have updated the text as well. Both the strength of the CMV promoter and the amount of DNA transfected were titrated and optimized. We compared endogenous ZAK expressed in WT HEK293T cells (Extended Data Fig. 1a, lanes 1-2) with the amount of ZAK expressed from our partial CMV promoter in ZAK KO HEK293T cells (Extended Data Fig. 1a, lanes 5-10) using a ZAK specific antibody (Fortis, A301-993A). We also compared ZAK activation (via JNK phosphorylation) between endogenous and transfected ZAK to select a condition where both expression level and activity were most closely matched (1.25 ug DNA). While included on the same blot (far right on blot), the complete promoter overexpression plasmid was not used for the evaluation of ZAK activity throughout the manuscript. Finally, we routinely checked ZAK levels in transfection experiments (especially with mutant analysis) to ensure that protein expression was equivalent.

5. In line 222, the authors state that loss of ribosome binding correlates with loss of ZAK activation and JNK phosphorylation. Is it because released ZAK cannot be autophosphorylated or this mutant cannot be autophosphorylated? I notice that in RACK1^{-/-} cells the endogenous ZAK protein is not phosphorylated when not bound to the ribosome. It requires a discussion.

Response:

For the typical inactive mutant that fails to bind ribosomes (e.g. pin, patch and RIH), we think that loss of ribosome binding is the explanation for the fact that these mutants fail to be activated. However, we can bypass this requirement for binding with a known constitutively active mutant (F368C). For example, the RIH mutant (Fig. 3e) completely loses ribosome binding and thus activation (similar to eS27-pin mutant); however,

ribosome dependent activation can be bypassed by combining the RIH mutant with a hyperactive SAM mutation (F368C) (Fig. 5h). These data indicate that the RIH variant ZAK is competent for autophosphorylation, if a mechanism for activation is introduced.

6. The importance of the study also lies in the fact that ZAK kinase activation determines cell fate. The authors have previously shown that ribotoxic stress drives UV-mediated cell death. Can the authors use the reconstituted cells with the different mutant ZAK kinases (including the pathogenic ones) and determine tolerance (by washing-off ANS) or vulnerability by using UV or by other means that the authors wish to show the physiological significance of the structural/mechanistic findings?

Response:

The Reviewer raises some ideas connected to recently published work from Sinha et al. 2024 indicating (1) that a tolerance regime is established under mild ribosome-collision inducing conditions where ZAK is activated then targeted for degradation and (2) that ZAK-mediated cell death is promoted downstream of more potent ribosome collision-inducing conditions. In light of these earlier studies, we would predict that cells expressing constitutively active ZAK (F368C and K387D) would be prone to pathway activation and cell death, even in the absence of collision-inducing treatments, and that the inactive ZAK variants would resist activation and cell death even in the presence of collision-inducing treatments.

Indeed, when the pathogenic variant F368C is characterized, we see little difference in pathway activation in the absence or presence of anisomycin. We did not include these data in the paper because a previously published paper had already characterized this mutant (Vind et al., Figure 6D). We observed similar trends with other hyperactive variants identified in our study as seen in Extended Data Fig. 8b, and chose one (K387D) for deeper analysis.

Redacted

More generally, we appreciate the reviewer request to characterize tolerance or cell death as a physiological read-out of our structural and biochemical studies. Here we used the ECHO Cellcyte1 to track cell confluency over time (as a proxy for cell death) since the tools (mutants etc) established in this study were not built into the reporter-carrying cell lines utilized in our earlier studies. There are several challenges here. First, because we are expressing via transient transfection, it is difficult to capture/compare early time points when proteins are just being expressed. Second, we know that the hyperactive mutants (F368C, K387D) are targeted for degradation even in the absence of ANS or UV unless they are expressed in the presence of the proteasome inhibitor MLN4924, which is toxic for long term growth experiments. Finally, we don't have the KTR reporters expressed that allow us to study pathway activation using microscopy. Nevertheless, we were able to follow cell confluency over 48 hours post-transfection. And, as expected, cells expressing F368C (the constitutive on mutation) appear to have slower growth rates (and more apoptosis based on floating cells) compared to cells expressing wildtype ZAK. While these data are consistent with the model, we do not feel that the signature is robust enough to warrant adding such experiments to the manuscript at this time. Moreover, we feel that the emphasis of this manuscript is really on structure and biochemistry, and we would strongly expect our hyper and hypo active variants to behave in a manner consistent with what we previously reported.

Redacted

Sinha, N. K. et al. The ribotoxic stress response drives UV-mediated cell death. *Cell* **187**, 3652–3670.e40 (2024).

Vind, A. C. et al. ZAK α Recognizes Stalled Ribosomes through Partially Redundant Sensor Domains. *Mol. Cell* **78**, 700–713.e7 (2020).

Overall, it was a pleasure to read this manuscript. I believe the suggested revisions will broaden the significance of the findings.

Referees' comments:

Referee #1 (Remarks to the Author):

The authors have addressed all my comments satisfactorily.

Referee #2 (Remarks to the Author):

The authors have addressed all of my concerns. I have no further suggestions and recommend the manuscript for publication.

Referee #3 (Remarks to the Author):

I co-reviewed this manuscript with one of the reviewers who provided the listed reports.

Referee #4 (Remarks to the Author):

I read the response to all reviewers and I found the revisions acceptable. The manuscript is suitable for publication in Nature.